# REPOAUDIT: An Autonomous LLM-Agent for Repository-Level Code Auditing

**Jinyao Guo**[* 1]  **Chengpeng Wang**[* 1]  **Xiangzhe Xu**[1]  **Zian Su**[1]  **Xiangyu Zhang**[1]

## Abstract

Code auditing is the process of reviewing code with the aim of identifying bugs. Large Language Models (LLMs) have demonstrated promising capabilities for this task without requiring compilation, while also supporting user-friendly customization. However, auditing a code repository with LLMs poses significant challenges: limited context windows and hallucinations can degrade the quality of bug reports, and analyzing large-scale repositories incurs substantial time and token costs, hindering efficiency and scalability.

This work introduces an LLM-based agent, RE-POAUDIT, designed to perform autonomous repository-level code auditing. Equipped with agent memory, REPOAUDIT explores the codebase on demand by analyzing data-flow facts along feasible program paths within individual functions. It further incorporates a validator module to mitigate hallucinations by verifying data-flow facts and checking the satisfiability of path conditions associated with potential bugs, thereby reducing false positives. REPOAUDIT detects 40 true bugs across 15 real-world benchmark projects with a precision of 78.43%, requiring on average only 0.44 hours and $2.54 per project.Also, it detects 185 new bugs in high-profile projects, among which 174 have been confirmed or fixed. We have open-sourced REPOAUDIT at https://github.com/PurCL/RepoAudit.

## 1. Introduction

The rapid innovation of large language models (LLMs) has remarkably enhanced the productivity of software developers (Wang et al., 2021; Rozière et al., 2023; Guo et al., 2024).

*Equal contribution [1]Department of Computer Science, Purdue University, West Lafayette, IN, USA. Correspondence to: Jinyao Guo <guo846@purdue.edu>, Chengpeng Wang <wang6590@purdue.edu>.

*Proceedings of the $42^{nd}$ International Conference on Machine Learning*, Vancouver, Canada. PMLR 267, 2025. Copyright 2025 by the author(s).

LLM-powered IDE plugins, such as Copilot, enable agile code generation by facilitating flexible interactions with LLMs (Barke et al., 2023). However, in the era of LLMs, auditing the rapidly expanding codebase presents a more formidable challenge than writing code. Traditional program analysis techniques, such as dynamic and static analysis, focus primarily on observing runtime behaviors or symbolically reasoning about intermediate code generated during compilation (King, 1976; Cadar et al., 2008; Calcagno et al., 2009; Sui and Xue, 2016; Shi et al., 2018a). Unfortunately, software systems are often non-executable and even uncompilable during the human-LLM collaborative development phase. Moreover, existing analysis techniques demand substantial expertise, such as a deep understanding of compiler internals (Zhang et al., 2024; Zhou et al., 2024). Therefore, current program analysis approaches often fall short in meeting the practical requirements of code auditing in real-world contexts (Johnson et al., 2013).

In recent years, extensive research has focused on leveraging LLMs for code auditing via prompt engineering (Fang et al., 2024; Hao et al., 2023; Ding et al., 2024). Unlike traditional analysis techniques, LLM-driven code auditing directly analyzes source code without program compilation or execution. By describing analysis requirements through natural language and few-shot examples in prompts, the customization of analysis is also significantly simplified. However, many existing LLM-driven code auditing techniques are largely restricted to small-scale codebases, such as smart contracts (Sun et al., 2024; Zhang and Zhang, 2024), and lack the capability to support repository-level code auditing in complex, real-world scenarios.

**Direct Prompting Hardly Works.** A straightforward solution is to break down a repository into smaller pieces and prompt the model with individual pieces. This approach often falls short for non-local bugs, which may require reasoning across a large number of interconnected code snippets spanning multiple functions, classes, and files. Even with significant future advancements in model reasoning capacity, the fundamental differences between programs and natural language texts for which transformer models were designed make LLMs insufficient for comprehensive repository analysis. In particular, a code repository can be conceptualized as an enormous graph, with nodes representing individual statements and edges capturing intricate

relationships such as control flow, data flow, and other interdependencies between statements. These relationships are precisely defined and immensely complex at the repository level. For instance, the number of data-flow edges in the project shadowsocks-libev exceeds one million, significantly surpassing the complexity of the implicit graphical structure present in any LLM's pre-training data sample. Such distribution differences render direct prompting ineffective as shown by our experiments in Section 2.2.

In addition, detecting many types of bugs requires reasoning about properties along specific program paths. For instance, a memory leak occurs when allocated memory is not freed along some program path. Detecting such so-called *path-sensitive* bugs (Shi et al., 2018a) necessitates unfolding the program's graph structure into individual paths and analyzing these paths one by one. However, this leads to the well-known *path explosion* problem, as the number of paths grows exponentially with the number of statements. Hence, direct prompting is akin to presenting a large project on an enormous screen and expecting a human auditor to identify bugs along complex and lengthy paths solely by reading and interpreting the code—a task highly unlikely to succeed.

**Human Auditing.** In practice, human auditors do not operate in such a manner. As revealed by existing cognitive science literature (Anicic et al., 2012), humans are highly effective at reasoning about events that occur in order, e.g., in temporal or spatial order. Human auditors hence tend to explore complex graphical structures in code by following paths that denote the execution order. To avoid excessive exploration, they traverse only a subset of paths most relevant to the targeted property, leveraging implicit abstraction to preclude irrelevant paths. An example of detecting null pointer dereference can be found in Section 2.1.

**Our Solution.** It is widely believed that LLMs operate similarly to humans but with significantly greater "endurance" and access to a broader spectrum of knowledge (Li et al., 2023; Long et al., 2024; Qian et al., 2024). Building on this perspective, we propose a novel LLM-based repository auditing agent, named REPOAUDIT, inspired by human auditing practices. REPOAUDIT addresses the fundamental misalignment between LLMs' tendency to reason sequentially and the inherently complex graphical structures of software repositories through path-sensitive and demand-driven graph traversal. By leveraging LLMs' abstraction capabilities, REPOAUDIT mitigates path explosion by excluding irrelevant code regions and sub-paths. Also, it minimizes inherent hallucinations by sanitizing final outputs through the validation of several well-formed properties.

More specifically, the agent REPOAUDIT consists of three components, including the initiator, the explorer, and the validator. First, the initiator identifies starting points based on the properties under investigation, such as NULL values when scanning for null pointer dereference bugs. Second, the explorer traverses the relevant functions on demand. Similar to how human auditors analyze code function by function, the explorer starts by querying the LLMs with the functions containing these starting points. Instead of explicitly and programmatically enumerating individual paths inside a function, as in traditional compiler-based automated scanners, the explorer leverages LLMs' inherent ability to implicitly distinguish relevant paths from irrelevant ones and only reasons about the former. If invocations to other functions and returns within a function are considered relevant after analyzing paths within the function—for instance, when null pointer values propagate through these function boundaries—the system synthesizes follow-up prompts to extend the scanning into callee or caller functions as needed. Third, REPOAUDIT checks the output of the explorer before storing it in the agent memory and also examines the bug report candidates by examining the path conditions of the buggy program paths. This validation design can significantly improve the precision of REPOAUDIT.

We implement REPOAUDIT powered with Claude 3.5 Sonnet and test it on three typical bug types of memory concurruption. We first evaluate REPOAUDIT upon fifteen real-world projects used in existing studies, with an average size of 251 KLoC. It is shown that REPOAUDIT effectively reproduces 21 previously reported bugs and uncovers 19 newly discovered bugs, 14 of which have already been fixed in the latest commit, achieving a precision of 78.43%. In contrast, the industrial static bug detector Amazon CODE-GURU reports 18 false positives (FPs) with no true positives (TPs), while Meta INFER also only reports seven TPs along with two FPs. Besides, REPOAUDIT exhibits high efficiency and incurs low token costs, averaging 0.44 hours and $2.54 per project. Powered by Deepseek R1, Claude 3.7 Sonnet, and OpenAI o3-mini, it achieves the precision of 88.46%, 86.79%, and 82.35%, respectively. To demonstrate its real-world impact, we further scan ten actively maintained repositories and detect 185 new bugs in two months, 95 and 79 of which have been confirmed and fixed by developers, respectively. Notably, REPOAUDIT facilitates development-time code auditing, which cannot be supported by Meta INFER and other compilation-dependent bug detectors. To the best of our knowledge, REPOAUDIT is the first purely LLM-driven code auditor for real-world code repositories.

## 2. Preliminaries

In this section, we first discuss the essence of repository-level code auditing. Next, we illustrate the limitations of the LLMs in this task. Finally, we highlight several intrinsic strengths of LLMs in tackling primitive tasks that can be leveraged to build our repository-level auditing tool.

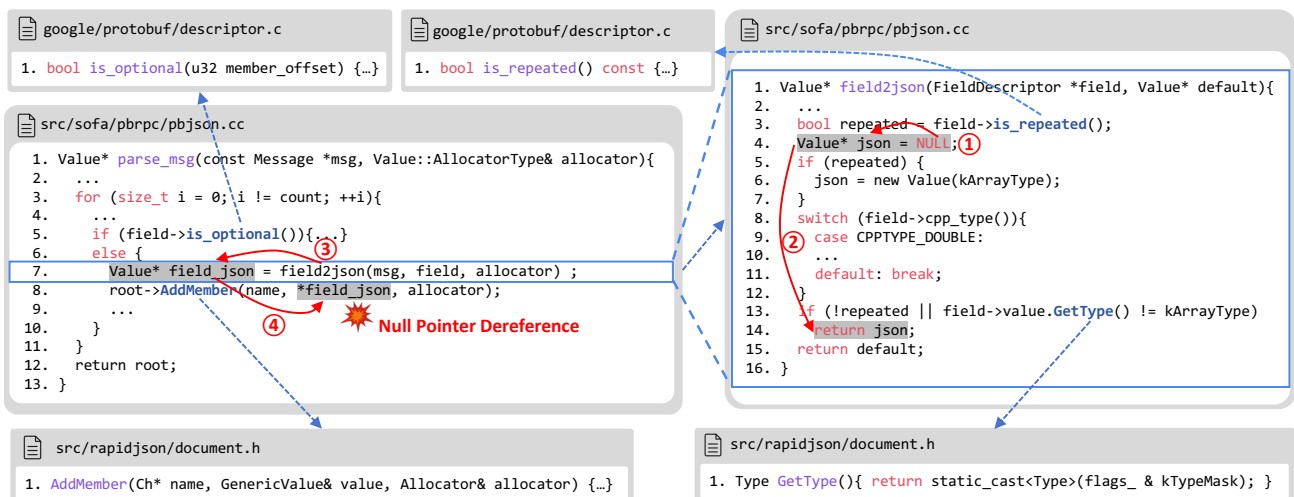

*Figure 1.* A simplified code snippet from the project `sofa-pbrpc` contains a real NPD bug found by REPOAUDIT. The blue dashed arrows indicate the edges in the call graph. The red solid arrows show the data-flow facts indicating the null value propagation. The call graph of the project contains 1,508 nodes and 6,196 edges, while its data dependence graph contains 160,875 nodes and 360,096 edges.

## 2.1. Auditing Entails Path-Sensitive Reasoning on Complex Graphs

While several bug types, such as API misuse (Li et al., 2021), require only localized reasoning upon abstract syntax trees (ASTs) and are relatively easy to detect with highly effective scanners, many critical bug types demand modeling the entire project as a massive graph and reasoning about properties along and across individual paths in that graph. For example, detecting Null Pointer Dereference (NPD) bugs relies on constructing and analyzing a specialized graph structure called the *data dependence graph* (DDG) (Ferrante et al., 1984), where the nodes represent statements, and the edges indicate *data-flow facts* between different program values at specific statements. Specifically, a data-flow fact exists from the variable $u$ at the statement $st_a$ to the variable $v$ at the statement $st_b$, denoted as $u@st_a \hookrightarrow v@st_b$, if the value of the variable $u$ at the statement $st_a$ may affect the value of the variable $v$ at the statement $st_b$ following some program path. A *program path* is a sequence of statements that follow the execution order. Also, it is *feasible* if there exists an input that satisfies all the conditional checks along the path. Hence, a data-flow fact can occur between two statements that are far apart, such as when a global variable is read and written across different directories.

Determining whether a data-flow fact may hold in the program requires collecting feasible program paths and analyzing data-flow facts along them. Consider the detection of Null Pointer Dereference (NPD) as an example. Given the DDG, the code auditor ought to find a chain of data-flow facts leading from a null value (as a *source* value) to a dereferenced pointer (as a *sink* value) along a feasible program path. In Figure 1, for instance, the function `field2json` initializes a null value at the line 4 of the function, inducing the data-flow fact $\text{NULL}@s_4 \hookrightarrow \text{json}@s_4$, which is labeled with ① in Figure 1. When `repeated` is false, the value of `json` at line 4 propagates to the return statement at line 14. This data-flow fact, denoted by $\text{json}@s_4 \hookrightarrow \text{json}@s_{14}$, is labeled with ②. Subsequently, in the function `parse_msg`, the return value of `field2json` is assigned to the pointer `field_json` at line 7 and further dereferenced at line 8 of the function `parse_msg`, which eventually causes an NPD bug.

Furthermore, we study the 2024 CWE Top 25 Most Dangerous Software Weaknesses, a curated list of the most critical and prevalent vulnerabilities underlying the 31,770 Common Vulnerabilities and Exposures (CVE) records reported in 2024. Our investigation reveals that 19 out of the 25 weakness (76%) categories necessitate *global, path-sensitive reasoning of* **source-sink reachability** *upon call graphs, DDGs, and control-flow graphs (CFGs)*, whereas only 6 categories can be effectively identified through localized analysis based on ASTs.

## 2.2. LLMs' Inadequacy

According to (Rozière et al., 2023; OpenAI, 2023), many foundation models are initially pre-trained on relatively short text or code snippets. For long contexts, models like DeepSeek-V3 (Liu et al., 2024), Llama3 series (Dubey et al., 2024), and QWen2.5-Coder (Hui et al., 2024) typically adopt NTK-aware length interpolation such as YaRN (Peng et al., 2024) to progressively extend context window from the initial 4K/8K to 128K tokens. Although these models perform well in "Needle in a Haystack" (Kamradt, 2023) evaluation, the task is intended for RAG-and does not align well with the path-sensitive program understanding ability needed in our application.

Please analyze the function *field2json* in Figure 1. We only care about the statements that affect the propagation of the NULL value at line 3. Please simplify the function *field2json* for program abstraction.

Here is the program after abstraction.

```
1.  Value* field2json(FieldDescriptor *field,
                        Value* default){
2.    bool repeated = field->is_repeated();
3.    Value* json = NULL;
4.    if (repeated) {
5.      json = new Value(rapidjson::kArrayType);
6.    }
7.    if (!repeated || field->value.GetType() != kArrayType)
8.      return json;
9.  }
```

(a) An example of program abstraction

What are the points-to facts of the variable *json* at line 8 in the function *field2json* shown in Figure 1?

The pointer variable *json* is initialized to NULL value at line 4 and then points to the newly allocated memory at line 6 if repeated is true. Hence, we have:
(1) repeated = true: The pointer variable *json* points to the object allocated at line 6;
(2) repeated = false: the pointer variable *json* is NULL

(b) An example of pointer handling

Collect the feasible program paths from the function *field2json* in Figure 1.

The branch conditions at lines 5 and 13 contradict.
Feasible path 1: Lines 1-5, 8-12, 13-14
Feasible path 2: Lines 1-5, 8-12, 13, 15
Feasible path 3: Lines 1-7, 8-12, 13-14

(c) An example of feasible program path exploration

*Figure 2.* Three examples of showing intrinsic strengths of LLMs

To validate our speculation, we conducted a controlled experiment where we prompt Claude 3.5 Sonnet with all the five functions shown in Figure 1 to identify the NPD bug, following a methodology similar to a recent study (Fang et al., 2024). This is a controlled experiment because, in practice, we cannot guarantee knowledge of the comprehensive set of functions associated with a bug. The model exhibited substantial hallucinations, reporting that almost all the dereferenced pointers have null values. Even if we improve the prompts by offering several few-shot examples and explanations on how a null pointer dereference bug occurs, the model still hallucinates, producing false positives and incorrect explanations. More numerical results can be found in Appendix A.

### 2.3. Intrinsic Strengths of LLMs

On the bright side, we observe that LLMs can effectively perform basic analyses when the scope is limited. This capability enables us to surpass traditional program analysis methods, which require compilation and struggle to scale efficiently. Specifically, we identify several primitive abilities critical for auditing: *program abstraction*, *pointer handling*, and *feasible program path exploration*. While traditional auditing tools rely on heavy-weight yet rigorous techniques to achieve these capabilities, skilled human auditors often rely on intuition to handle such challenges within a constrained analysis scope. We observe similar traits in LLMs.

#### 2.3.1. PROGRAM ABSTRACTION

Abstraction is essential for scalability in program analysis. Given a property, a set of initial program points, and a defined scope (e.g., a function), abstraction identifies the subset of statements relevant to the property within the scope. These statements form a self-contained, smaller program, significantly reducing the number of paths to analyze. For instance, in the NPD detection (as illustrated in Section 2.1), abstraction targets how null values are propagated

in the program and focuses on key statements such as null pointer assignments, value propagations, conditional checks guarding the propagations, and cross-function propagations (e.g., passing the pointer to a callee function via a parameter, returning it to a caller function, or writing it to a global variable). In a preliminary experiment shown in Figure 2(a), for example, we feed the function `field2json` in Figure 1 to Claude 3.5 Sonnet and ask it to abstract the program. Observe that the program returned by the LLM only contains critical statements relating to the null value propagation, while irrelevant ones are removed, such as the switch statement from line 8 to line 12 in the function `field2json` in Figure 1. Notably, human auditors often perform such abstraction implicitly, enabling them to analyze complex code without the excessive path exploration that hampers classic tools like symbolic execution (Cadar et al., 2008) and software model checking (Clarke, 1997).

#### 2.3.2. POINTER HANDLING

Pointer variables in C and reference variables in Java may point to different memory objects depending on their run-time values. Consequently, reads and writes via pointer dereferences create data-flow facts that dynamically vary based on pointer values. Determining the set of memory objects a pointer variable may point to is known as *points-to analysis* (Smaragdakis et al., 2015), one of the most challenging problems for downstream static analysis, such as DDG construction. Traditional points-to analysis techniques often rely on conservative and relational methods, which tend to significantly over-approximate the set of possible memory objects, complicating downstream analysis tasks. In contrast, human auditors can often intuitively and accurately determine the points-to facts of a specific pointer variable through their understanding of program semantics. Advanced LLMs exhibit similar capabilities, particularly within the scope of individual functions. As shown in Figure 2(b), for example, we prompt Claude 3.5 Sonnet with the function `field2json` in Figure 1 and query the points-to

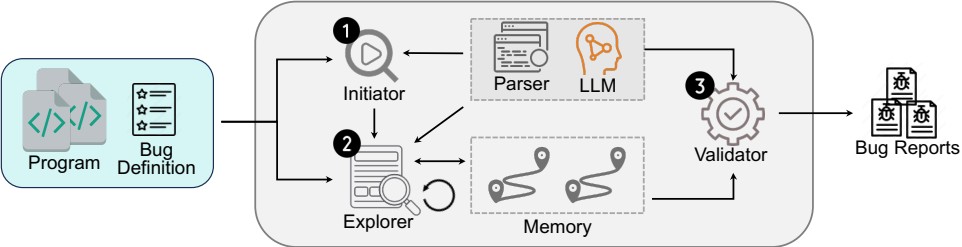

Figure 3. The architecture of REPOAUDIT

facts of the return value of the function. The model accurately identifies the two possible points-to facts within the function. It even identifies the path constraints that make the corresponding points-to facts hold. In contrast, the symbolic static analyzer SVF (Sui and Xue, 2016) computes the points-to facts without path conditions by default due to its inherent limitations in semantic analysis.

### 2.3.3. FEASIBLE PROGRAM PATH EXPLORATION

A bug report is often considered valid only if it provides a feasible program path (from the root cause to the symptom) as evidence. To determine feasibility, traditional tools rely on modeling conditional checks along a path as symbolic constraints (e.g., in the form of first-order logic formulas) and querying a theorem prover (e.g., SMT solver Z3 (de Moura and Bjørner, 2008)) to check if an input can satisfy the constraints. This process is computationally expensive and prone to failure, as converting program paths to logic formulas requires exploring an explosively large number of program paths and modeling a wide range of program behaviors that lack direct representation in logical term, such as loops, array indexing, aliasing, pointer arithmetic, and unbounded string operations. In contrast, humans, as well as the LLMs, rely on abstraction and intuitive logical reasoning to assess feasibility, which is highly effective within a limited scope. In Figure 2(c), we demonstrate an example of feasible program path exploration upon the function `field2json` using Claude 3.5 Sonnet. Observe that the model can skip the irrelevant statements, such as the switch statement, avoiding exploring a huge number of program paths. It also discovers the contradiction between the branch conditions at lines 5 and 13, which refutes program paths covering both lines 6 and 14, thereby reporting the three feasible program paths in Figure 2(c).

## 3. REPOAUDIT

Building on the findings of the previous section, the core design rationale of REPOAUDIT is as follows: Since LLMs struggle to reason about path-sensitive properties in large-scale program graphs, REPOAUDIT employs an agent-centric approach to navigate these graphs externally, prompting the model with one unit (e.g., a function) at a time. This demand-driven navigation adapts based on the model's re-

sponses for each function, and a dedicated agent memory ensures that analysis results across functions are seamlessly shared. To achieve cost-effective and path-sensitive analysis at the function level, REPOAUDIT leverages LLMs' intrinsic capabilities by providing explicit prompts for program abstraction, pointer handling, and feasible program path exploration, fully capitalizing on the model's strengths.

Figure 3 depicts the architecture of our agent. The *initiator* tool (❶) takes a specified bug definition and the target repository as inputs, identifying the source values for analysis, such as the null values for the NPD detection. Each source value triggers a scanning procedure. The *explorer* tool (❷) conducts iterative, demand-driven exploration of the repository by prompting the model to analyze one function at a time. The results of each analysis are stored in the agent memory, guiding further exploration. The analysis prompts are dynamically generated for each function, providing detailed instructions for abstraction tailored to the specific function and adjusted based on the results of prior analyses. While the agent may still hallucinate in the analysis of single functions and produce false positive bug reports, a set of *validator* tools (❸) verify the result of the explorer from multiple perspectives, including the validity of the control flow order and the satisfiability of the path conditions across functions.

### 3.1. Initiator

The initiator identifies the starting points of scanning (i.e., the sources). In our implementation, we employ the `tree-sitter` parsing library to create a suite of tailored pattern matchers for the sources of the bug types supported by REPOAUDIT. These matchers are concise, often consisting of just a few lines of code, and require a one-time implementation effort. Alternatively, recent advancements offer a promising avenue of synthesizing such matchers using LLMs (Wang et al., 2024a).

### 3.2. Explorer

For each source value identified by the initiator, the explorer conducts a round of scanning over the repository. During each round, the explorer traverses a subset of functions on demand, beginning with the identified sources. It queries the LLM to analyze one function at a time, storing the results

**Task:** Determine data-flow facts starting from a given value along different program paths in a single function.

**Hints:** You can collect the feasible program paths into 3 **steps**:
(1) Identify the aliased pointers in the program.
(2) Extract the critical statements relating to the value propagation, like function call statements, return statements, and assignments.
(3) Collect the feasible program paths covering critical statements. For each feasible program path, simulate the program execution and determine the data-flow facts starting from the given value.

**Examples:** Here are several examples.
Example 1: User: [Program] [Question]
System: <Path information: Lines 4-7, 11> <Explanation>
<Data-flow facts: $ptr@\ell_4 \hookrightarrow qtr@\ell_7, ptr@\ell_4 \hookrightarrow rtr@\ell_{11}$ >
[Other examples]

**Question**: Given [FUNCTION], what are the data-flow facts from [VAL_NAME] at line [VAL_LINE] along different program paths?

*Figure 4.* The prompt template for analyzing individual functions

in the agent memory. The explorer performs several actions, each guided by corresponding prompts. These actions include: *analyzing individual functions*, *selecting functions for exploration*, and *generating bug report candidates*. We will introduce the details of the three actions as follows.

### 3.2.1. ANALYZING INDIVIDUAL FUNCTIONS

As discussed in Section 2.1, scanning for most bug types can be reduced to traversing a limited set of graphs, such as the DDG and the CFG. This enables the use of general analysis prompts for these graph types, eliminating the need for bug-specific prompts. Moreover, instead of explicitly enumerating and analyzing individual paths, we leverage the intrinsic capabilities of LLMs to distinguish paths and perform path-sensitive reasoning. The key of this approach lies in pointer handling and program abstraction demonstrated in Section 2.3.2 and Section 2.3.1, respectively, which implicitly reduces the function to a significantly smaller code snippet with far fewer paths. This can eventually facilitate the efficient exploration of feasible program paths demonstrated in Section 2.3.3. We design the prompt template used for analyzing individual functions in Figure 4. After describing the task at the beginning, we offer the three-step hints to the LLM, enforcing the LLM to unleash its power step-by-step in pointer handling, abstraction, and feasible program path exploration. By offering several few-shot examples, we pose the question at the end and ask the LLM to identify the data-flow facts starting from the initial value(s) of interest along different feasible program paths.

Consider the function `filed2json` and the initial null value at the line 4 as an example. Utilizing the instrinsic strengths of LLMs, REPOAUDIT collects three feasible program paths shown in Figure 2(c). By simulating the program execution along the first program path, denoted by $p_1$, the LLM identifies the data-flow facts $\text{NULL}@s_4 \hookrightarrow \text{json}@s_4$ and $\text{json}@s_4 \hookrightarrow \text{json}@s_{14}$. For the second and third program paths, denoted by $p_2$ and $p_3$, the LLM only identifies the data-flow fact $\text{NULL}@s_4 \hookrightarrow \text{json}@s_4$.

**Agent Memory.** After analyzing a function, the explorer obtains a set of data-flow facts for each feasible program path and stores them into the memory of the agent. Specifically, the agent memory is a function $\mathcal{M}$ relating to the function $f$ and a program value $v@s$. Each element in $\mathcal{M}(f, v@s)$ is a pair of a program path and a set of data-flow facts. After analyzing the function `field2json` and the null value at the line 4, for instance, the memory maps $(\text{field2json}, \text{NULL}@s_4)$ as follows:

$$\mathcal{M}\big(\text{field2json}, \text{NULL}@s_4\big)$$
$$= \Big\{ \big(p_1, \{\text{NULL}@s_4 \hookrightarrow \text{json}@s_4, \text{json}@s_4 \hookrightarrow \text{json}@s_{14}\}\big),$$
$$\big(p_2, \{\text{NULL}@s_4 \hookrightarrow \text{json}@s_4\}\big),$$
$$\big(p_3, \{\text{NULL}@s_4 \hookrightarrow \text{json}@s_4\}\big)\Big\}$$

### 3.2.2. SELECTING FUNCTIONS FOR EXPLORATION

After analyzing a function, if the targeted program value propagates across function boundaries, the explorer queries the underlying call graph to identify the relevant functions for further exploration. This follow-up exploration is guided by the program values escaping the current function boundaries. For example, in Figure 1, the null value at the line 4 of `field2json` is propagated to the return value. Hence, in the next step, the explorer analyzes the caller function of `field2json`, i.e., `parse_msg`, and examines how the return value is propagated.

Note that if the program value does not escape, the analysis does not lead to the explorations of other functions. For example, if we consider the second and third program paths, i.e., $p_2$ and $p_3$ in the example shown in Section 3.2.1, the value of `json` at the line 4 of the function `field2json` does not propagate. Hence, the explorer does not enter any other functions. Furthermore, we leverage the existing data-flow facts stored in the memory as caches to avoid redundant analysis. Specifically, before analyzing the propagation of a specific program value in a function, the explorer first checks the agent memory and determines whether this has been done before. In our evaluation, we will quantify how the caching strategy reduces computational costs.

### 3.2.3. GENERATING BUG REPORT CANDIDATES

After analyzing a function, the explorer evaluates whether any new bug candidates can be identified. Consider the NPD detection as an example. The explorer examines whether it identifies any new data-flow facts reaching a sink value, i.e., a dereferenced pointer. If so, a bug report is generated by assembling the complete trail of data-flow facts across functions along with the corresponding inter-procedural program path. For example, the explorer enters the function `parse_msg` and examines how the return value of `field2json` propagates. Based on the two discovered data-flow facts labeled as ③ and ④ in Figure 1, the explorer

| | |
|---|---|
| **Task:** Determine whether a given path is feasible or not. | |
| **Hint:** If branch conditions conflict or variable values contradict the required conditions, the path is infeasible. | |
| **Examples:** Here are several examples. Example 1: User: [Program] [Question] System: <Explanation> <Answer: Yes or No> [Other examples] | |
| **Question**: Given the functions [FUNCTION], is the following program path feasible? [PATH] | |

*Figure 5.* The prompt template for feasibility validation

reaches a sink value, i.e., the dereferenced pointer at line 8. Hence, the explorer identifies a potential bug and report it by concatenation the data-flow facts ①, ②, ③, and ④. For several bug types, such as MLK, the explorer reports a potential bug if it fails to reach any sink point along feasible paths, e.g., the argument of `free` function.

### 3.3. Validator

To improve the quality of bug reports, we introduce two kinds of validation mechanism for the explorer.

**Alignment Validation of Data-flow Facts and Control Flow.** When dealing with complex code, the LLM may hallucinate and incorrectly infer a data-flow fact $u@s_1 \hookrightarrow v@s_2$ along some program path $p$ that violates the control-flow order. Specifically, this implies that a statement $s_2$ must appear after another statement $s_1$ in the program path $p$, yet the model concludes that a variable defined at $s_2$ can be used by $s_1$. To detect such misalignments, a parsing-based analyzer is employed to verify the control-flow order. Only the data-flow facts that conform to the control-flow order will be stored in the agent memory.

**Path Feasibility Validation.** While path feasibility within individual functions is inherently checked by the explorer, contradictions can arise in conditional checks across functions, resulting in infeasible inter-procedural program paths. Recall that each bug report is the concatenation of multiple data-flow facts, such as ①, ②, ③, and ④ in Figure 1. The data-flow facts in different functions may hold under the specific conditions along the corresponding program paths. A bug report candidate is valid only if the path conditions in different functions do not contradict, implying their logical conjunction satisfiable. To check the validity, we prompt the LLM with the inter-procedural path and task it with identifying any contradictions in the path conditions. If a contradiction is found, the bug report is discarded. The prompt template is shown in Figure 5.

## 4. Evaluation

We utilize the `tree-sitter` parsing library to provide a set of primitive tools for REPOAUDIT, e.g., call graph constructor and control flow order validator. We select LLM Claude 3.5 Sonnet to power REPOAUDIT. Following the

*Table 1.* The statistics of evaluation subjects

| Bug Type | ID | Repository Name | Size (LoC) | Stars |
|---|---|---|---|---|
| NPD | N1 | sofa-pbrpc | 40,723 | 2.1K |
| | N2 | ImageMagick/MagickCore | 242,555 | 12.6K |
| | N3 | coturn/src/server | 8,976 | 11.7K |
| | N4 | libfreenect | 37,582 | 3.6K |
| | N5 | openldap | 442,955 | 486 |
| MLK | M1 | libsass | 40,934 | 4.3K |
| | M2 | memcached | 14,654 | 13.7K |
| | M3 | linux/driver/net | 914,025 | 186K |
| | M4 | linux/sound | 1,378,262 | 186K |
| | M5 | linux/mm | 171,721 | 186K |
| UAF | U1 | Redis | 179,723 | 67.7K |
| | U2 | linux/drivers/peci | 2,130 | 186K |
| | U3 | shadowsocks-libev | 71,080 | 186K |
| | U4 | wabt-tool | 3,214 | 7K |
| | U5 | icu/icu4c/source/i18n | 220,359 | 2.9K |

common practice in evaluating reasoning tasks (Ye et al., 2023), we set the temperature to 0.0 to reduce the randomness. Similar to existing code auditing works (Heo et al., 2017), we introduce an upper bound $K$ on the calling context and set it to 4, i.e., REPOAUDIT investigates data-flow facts across a maximum of four functions.

### 4.1. Bug Types and Dataset

We focus on three bug types, namely NPD, MLK, and UAF, which are among the CWE Top 25 most dangerous weaknesses. Our evaluation first aims to reproduce bugs reported in previous works. Specifically, we investigate recent works in the venues of computer security and software engineering and collect the bug reports published by the authors (Huang et al., 2024; Shi et al., 2021; 2018b). Also, we attempt to detect new bugs within the targeted code repositories. As shown by Table 1, we choose five well-maintained projects for each bug type from the bug reports of previous works, which mostly have thousands of stars on GitHub with 251 KLoC and thousands of functions on average.

### 4.2. Evaluation Results

**Main Result.** As shown in Table 2, REPOAUDIT successfully detects all the previously reported bugs, and meanwhile, reports 19 new bugs in historic versions, 14 of which have already been fixed in the latest commit. In total, it reports 40 TPs and 11 FPs, resulting in a precision of 78.43%. Notably, 21 TPs are inter-procedural bugs. REPOAUDIT also demonstrates both efficiency and cost-effectiveness. On average, it takes only 0.44 hours (i.e., 1,577.22 secs) to analyze a project, completing the code auditing within 100.67 prompting rounds. The average cost per project audit is $2.54, with each true bug detection costing $0.95. Hence, REPOAUDIT can effectively detect the bugs upon real-world programs with low time cost and financial cost.

**Comparison with LLM-driven Bug Detectors.** We compare REPOAUDIT with two kinds of LLM-driven techniques, namely end-to-end few-shot CoT prompting-based approaches (Chen et al., 2023; Ding et al., 2024) and agent-

*Table 2.* The statistics of REPOAUDIT powered by Claude 3.5 Sonnet. The column "Old" denotes the bugs reported by existing works. A pair $(m, n)$ in the column **New** indicates $m$ new bugs detected in the versions used by existing works, $n$ of which still exist in the latest versions. The columns "Intra" and "Inter" show the numbers of intra-procedural and inter-procedural bugs, respectively.

| Bug Type | ID | TP | | FP | Feature | | # Prompts | # Tokens | | Financial ($) | Time (s) |
|---|---|---|---|---|---|---|---|---|---|---|---|
| | | Old | New | | # Intra | # Inter | | Input | Output | | |
| NPD | N1 | 1 | (3,3) | 2 | 0 | 4 | 145 | 709,919 | 55,863 | 2.97 | 2026.13 |
| | N2 | 7 | (1,0) | 0 | 4 | 4 | 17 | 97,717 | 8,518 | 0.42 | 283.84 |
| | N3 | 1 | (1,0) | 3 | 1 | 1 | 109 | 599,674 | 52,936 | 2.59 | 1747.90 |
| | N4 | 1 | (0,0) | 1 | 0 | 1 | 29 | 126,852 | 13,654 | 0.59 | 435.09 |
| | N5 | 1 | (5,4) | 1 | 0 | 6 | 63 | 420,710 | 31,375 | 1.73 | 1059.57 |
| MLK | M1 | 1 | (2,1) | 1 | 2 | 1 | 205 | 1,132,763 | 85,279 | 4.68 | 2,917.91 |
| | M2 | 1 | (6,6) | 2 | 4 | 3 | 146 | 845,148 | 71,243 | 3.60 | 2282.31 |
| | M3 | 1 | (0,0) | 0 | 1 | 0 | 2 | 10,481 | 1,019 | 0.05 | 34.34 |
| | M4 | 1 | (0,0) | 0 | 1 | 0 | 1 | 5691 | 619 | 0.03 | 17.94 |
| | M5 | 1 | (0,0) | 0 | 1 | 0 | 35 | 181,348 | 20,779 | 0.86 | 599.92 |
| UAF | U1 | 1 | (0,0) | 0 | 1 | 0 | 36 | 179,939 | 17,547 | 0.80 | 582.23 |
| | U2 | 1 | (0,0) | 0 | 1 | 0 | 2 | 8900 | 869 | 0.04 | 31.95 |
| | U3 | 1 | (0,0) | 0 | 1 | 0 | 48 | 317,713 | 23,067 | 1.30 | 791.98 |
| | U4 | 1 | (0,0) | 0 | 1 | 0 | 10 | 48,087 | 5,883 | 0.23 | 185.22 |
| | U5 | 1 | (1,0) | 1 | 1 | 1 | 662 | 4,534,444 | 303,645 | 18.15 | 10,661.98 |
| **Average** | | | | | | | **100.67** | **614,625.73** | **46,153.07** | **2.54** | **1,577.22** |

*Table 3.* The statistics of Meta INFER and Amazon CODEGURU

| Bug Type | ID | Build | INFER | | CODEGURU | |
|---|---|---|---|---|---|---|
| | | | TP | FP | TP | FP |
| NPD | N1 | ✗ | N/A | N/A | 0 | 0 |
| | N2 | ✓ | 1 | 0 | 0 | 0 |
| | N3 | ✓ | N/A | N/A | 0 | 0 |
| | N4 | ✓ | 5 | 0 | 0 | 2 |
| | N5 | ✓ | 1 | 2 | 0 | 3 |
| MLK | M1 | ✓ | 0 | 0 | | |
| | M2 | ✓ | 0 | 0 | | |
| | M3 | ✓ | N/A | N/A | N/A | N/A |
| | M4 | ✓ | 0 | 0 | | |
| | M5 | ✓ | N/A | N/A | | |
| UAF | U1 | ✓ | N/A | N/A | 0 | 0 |
| | U2 | ✓ | N/A | N/A | 0 | 0 |
| | U3 | ✗ | N/A | N/A | 0 | 0 |
| | U4 | ✓ | 0 | 0 | 0 | 0 |
| | U5 | ✓ | 0 | 0 | 0 | 13 |
| **Total** | | | **7** | **2** | **0** | **18** |

centric approaches (Wang et al., 2024a; Li et al., 2024a;b). It is shown that CoT prompting can only detect one true bug in the single function-level bug detection and 10 true bugs in the multiple-level bug detection when analyzing the functions relating to the true bugs detected by REPOAUDIT. For the agent-centric solution LLMDFA, the numbers of its prompting rounds and token costs are 165.23 and 123.18 times with the ones of REPOAUDIT on average when it analyzes relevant functions relating to the buggy program paths, showing its high computation costs in analyzing real-world programs. More details can be found in Appendix A.

**Comparison with Industrial Tools.** We compare REPOAUDIT with two representative industrial tools, namely Meta INFER (Meta, 2025) and Amazon CODEGURU (Amazon, 2025). As shown in Table 3, two projects (labeled by ✗) cannot be successfully built in our environment. Another five projects (labeled with NA) cannot be handled by INFER due to incompatibilities, a prominent limitation of build/compilation dependent tools. In total, Meta INFER reports seven

true bugs and two false positives. Amazon CODEGURU supports the detection of NPD and UAF bugs, while it reports 18 false positives with no true positives. Compared with Meta INFER, REPOAUDIT obtains comparable precision while detecting more true bugs. More detailed illustrations on the comparison results are provided in Appendix B.

**Ablation Study.** We introduce three ablation variants of REPOAUDIT, without abstraction, without validators, and without caching. We observe that without abstraction, the number of TPs is decreased by 47.50% and the number of FPs is increased by 181.82%. Without validators, the number of FPs increases by 245.45%. Without caching, the costs become 3-4 times higher on average, and in the worst case, 30 times higher. Details can be found in Appendix C.

**Different Model Choices and Temperature Settings.** We also evaluate REPOAUDIT with Deepseek R1, Claude 3.7 Sonnet, and OpenAI o3-mini, achieving precisions of 88.46%, 86.79%, and 82.35%, respectively. In addition, we assess REPOAUDIT powered by Claude 3.5 Sonnet under the temperatures of 0.25, 0.5, 0.75, and 1.0, achieving consistently high precision ($\geq 72.92\%$) and recall ($\geq 85.71\%$). More details are provided in Appendix D and Appendix E.

**Real-World Impact.** We further scan nine open-source GitHub projects spanning diverse domains and sizes ranging from 14K to 1.7M LoC. On average, they have 420K LoC and 8.8K GitHub stars, reflecting their complexity and popularity. Table 4 shows the detailed statistics. Specifically, the columns **TP** and **FP** show the numbers of TPs and FPs reported by REPOAUDIT, respectively, while the columns **Con** and **Fix** denotes the numbers of confirmed bugs and fixed bugs, respectively. Overall, REPOAUDIT detects 185 true bugs with the precision of 85.71%. Notably, 95 and 79 bugs confirmed and fixed by developers, respectively.

Table 4. The statistics of REPOAUDIT in nine additional real-world projects

| Project | Size (LoC) | Stars | NPD | | | | MLK | | | | UAF | | | |
|---|---|---|---|---|---|---|---|---|---|---|---|---|---|---|
| | | | TP | FP | Con | Fix | TP | FP | Con | Fix | TP | FP | Con | Fix |
| clickhouse-odbc | 209,197 | 258 | 0 | 0 | 0 | 0 | 2 | 0 | 0 | 2 | 0 | 1 | 0 | 0 |
| htop | 35,910 | 6.9K | 1 | 3 | 0 | 1 | 0 | 2 | 0 | 0 | 0 | 0 | 0 | 0 |
| TrinityEmulator | 1,767,100 | 292 | 2 | 2 | 0 | 2 | 4 | 0 | 0 | 4 | 0 | 0 | 0 | 0 |
| rtl_433 | 66,253 | 6.5K | 0 | 0 | 0 | 0 | 1 | 1 | 0 | 1 | 0 | 0 | 0 | 0 |
| frr | 1,050,020 | 3.6K | 4 | 5 | 4 | 0 | 0 | 1 | 0 | 0 | 0 | 0 | 0 | 0 |
| libuv | 80,360 | 25K | 86 | 2 | 84 | 2 | 1 | 2 | 0 | 1 | 0 | 0 | 0 | 0 |
| openldap | 442,955 | 486 | 8 | 2 | 0 | 8 | 0 | 0 | 0 | 0 | 3 | 0 | 0 | 0 |
| nginx | 505,513 | 26.4K | 5 | 2 | 0 | 0 | 0 | 0 | 0 | 0 | 0 | 0 | 0 | 0 |
| memcached | 14,654 | 13.7K | 1 | 3 | 1 | 0 | 67 | 4 | 6 | 58 | 0 | 0 | 0 | 0 |
| Total | | | 107 | 19 | 89 | 13 | 76 | 10 | 6 | 66 | 3 | 2 | 0 | 0 |

## 4.3. Limitations and Future Works

Apart from the false positives and negatives caused by LLM hallucinations, REPOAUDIT faces several limitations. First, the analysis overhead of REPOAUDIT is highly sensitive to the number of source elements within a repository. In cases where a large number of sources exist, the tool may incur substantial time and token costs. Second, REPOAUDIT is not sound in detecting inter-procedural data-flow facts, as it currently limits flow analysis to a maximum of four functions. This constraint may cause it to miss complex bugs involving longer call chains. To address these limitations, we outline several directions for future improvement:

**Enhancing Model Reasoning Capabilities:** We can improve the LLM's ability to reason by either adopting more advanced models or fine-tuning the current models for specific tasks. Fine-tuning can be tailored to particular sub-tasks like exploring feasible program paths in single functions. Additionally, incorporating specialized training datasets focused on programming languages, debugging, and code analysis could further enhance the model's accuracy in these contexts.

**Expanding the Tool Suite for Better Retrieval:** Developing a more comprehensive tool suite that facilitates the retrieval across the entire codebase is crucial. A significant enhancement would involve integrating existing compilation-free analysis tools to identify all potential branches and loops within a program. This integration would offer a clearer representation of the program's control flow structure to the model. Such an improved RAG design has the potential to significantly reduce the false positives and false negatives produced by RepoAudit.

## 5. Related Work

A considerable volume of literature has focused on utilizing LLMs for code auditing (Zheng et al., 2025; Zheng et al.; Li et al., 2024b;a; Wang et al., 2024a). In recent years, various benchmarks like BigVul (Fan et al., 2020), PrimeVul (Ding et al., 2024), and DiverseVul (Chen et al., 2023) have been established. Nevertheless, these benchmarks lack calling context for the buggy functions, thereby degrading the validity of such function-level code auditing techniques (Risse and Böhme, 2024). As for repository-level code auditing, existing techniques can generally be categorized into two groups. The first group harnesses LLMs to provide specialized pre-knowledge to symbolic code analyzers or examine initial bug reports, while the main body of the code base is scanned by conventional symbolic analyzers (Wang et al., 2024b; Li et al., 2024b;a). Typically, IRIS employs LLMs to pinpoint sensitive values within programs (Li et al., 2024b), aiding traditional analyzers like CODEQL (Avgustinov et al., 2016) in taint-style bug detection. However, due to the compilation reliance on symbolic analysis, these techniques lack the capacity for IDE-time analysis. The second category utilizes LLMs as code interpreters, extracting semantic properties via prompt engineering (Fang et al., 2024; Wang et al., 2024c;a; Sun et al., 2024). Typically, LLMDFA (Wang et al., 2024a) and LLMSAN (Wang et al., 2024c) incorporate few-shot CoT prompting to discover data-flow paths for bug detection. REPOAUDIT belongs to the latter category. Unlike LLMDFA, REPOAUDIT adopts a demand-driven strategy for codebase exploration, which avoids exhaustive data-flow summary generation for all functions, thereby enhancing analysis scalability. Additionally, REPOAUDIT outperforms LLMSAN in terms of recall by employing individual function analysis instead of repository-level end-to-end prompting, effectively mitigating hallucinations in the overall analysis.

## 6. Conclusion

This paper introduces REPOAUDIT, an autonomous LLM-agent that facilitates precise and efficient repository-level code auditing. By mimicking manual code auditing, RE-POAUDIT leverages the intrinsic strength of the LLM, such as program abstraction, and conducts the path-sensitive reasoning. Powered by Claude-3.5-Sonnet, it detects 40 true bugs in 15 real-world benchmark projects, achieving the precision of 78.43% and reproducing all the bugs discovered by existing techniques. It also detects 185 previously unknown bugs in nine high-profile open-source projects, 174 of which have been confirmed or fixed by developers.

## Acknowledgement

We are grateful to the Center for AI Safety for providing computational resources. This work was funded in part by the National Science Foundation (NSF) Awards SHF-1901242, SHF-1910300, Proto-OKN 2333736, IIS-2416835, DARPA VSPELLS - HR001120S0058, ONR N00014-23-1-2081, and Amazon. Any opinions, findings and conclusions or recommendations expressed in this material are those of the authors and do not necessarily reflect the views of the sponsors.

## Impact Statement

This paper presents work whose goal is to advance the field of Machine Learning, targeting a complicated code-reasoning task, namely repo-level code auditing. We demonstrate the limitations of our work above. We do not expect our work to have a negative broader impact, though leveraging LLMs for repo-level code auditing may come with certain risks, e.g., the leakage of source code in private organizations and potential high token costs. Meanwhile, it is worth more discussions to highlight that our work has the potential to dramatically change the field of software engineering with the power of LLMs. Specifically, LLM-powered code auditing not only enables the analysis of incomplete programs with little customization but addresses other challenges in the code auditing.

First, classical code auditors primarily rely on specific versions of intermediate representations (IRs) generated by compilation infrastructures. As compilation infrastructures evolve, IR formats often change across compiler versions, necessitating continuous adaptation of the analysis implementation. For instance, the Clang compiler has experienced ten major version updates over the past decade, each introducing variations in the generated IR. These differences require substantial manual effort to migrate and maintain compatibility with newer IRs. In contrast, LLM-powered code auditing operates directly on source code, inherently supporting multiple language standards and eliminating the dependency on compiler-specific IRs.

Second, classical code auditing heavily relies on various abstraction designs, particularly in pointer analysis, which serves as a foundational pre-analysis step. Developers must carefully select and implement specific analysis strategies—such as Andersen-style or Steensgaard's pointer analysis—each involving distinct trade-offs in precision and scalability. This process requires substantial implementation effort and domain expertise. In contrast, LLMs, which are inherently aligned with program semantics, can interpret code behavior directly. As a result, they obviate the need for manually crafting abstractions or implementing analysis algorithms tailored to particular precision levels.

Third, classical code auditing requires reasoning about the semantics of IRs and reimplementing the same algorithm for different languages. In contrast, LLMs serve as general code interpreters and have exceptional performance in understanding short code snippets, no matter which programming languages are used. By following similar prompting strategies, we can easily extend REPOAUDIT to analyze programs in other languages, including but not limited to C/C++, Python, and JavaScript.

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

## A. Comparison with LLM-driven Detectors

**Setup and Metrics.** To the best of our knowledge, there are two kinds of LLM-driven code auditing techniques, namely end-to-end few-shot chain-of-thought (CoT) prompting-based approaches and agent-centric approaches. Specifically, the former can be divided into two categories, including single-function level bug detection and multiple-function level bug detection. First, the single-function level

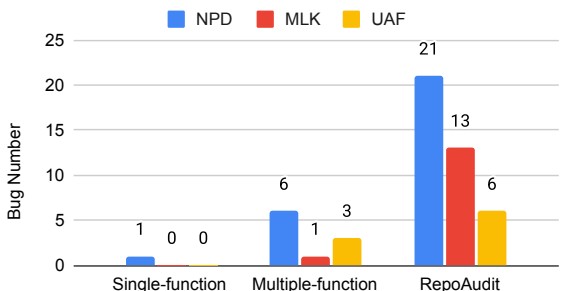

*Figure 6.* The comparison results with single-function level and multiple-function level bug detectors

bug detection is widely adopted and evaluated by many recent studies (Chen et al., 2023; Ding et al., 2024). These techniques are applicable for models with a limited context length. To compare with single-function level bug detection, we collect all the functions that contain the sink values yielding TPs to the LLM along with few-shot examples, asking the LLM to determine whether the function can introduce specific types of bugs. Second, multiple-function level bug detectors attempt to feed the whole program to the LLM so that the calling contexts of buggy functions can be included (Wang et al., 2024c). Unfortunately, the huge size of a real-world software system often makes the prompts exceed the context limit of LLM. Hence, we only feed the relevant functions covered by the buggy program paths to the LLM in our evaluation.

Among agent-centric approaches (Wang et al., 2024a; Li et al., 2024a;b), we select LLMDFA (Wang et al., 2024a) as a baseline for comparison, as it supports compilation-free and customizable analysis. Since LLMDFA cannot support the MLK detection, we focus our comparison between REPOAUDIT and LLMDFA specifically on NPD and UAF bugs. LLMDFA works by summarizing all data-flow facts for each function and then correlating these facts. Its overall computational costs—time and token costs—can be substantial. To avoid the excessive computation costs, we conduct a group of controlled experiments under two settings. In the first setting, LLMDFA is only applied to functions covered by buggy program paths, generating data-flow summaries for these functions. In the second setting, LLMDFA is tasked with generating data-flow summaries for functions reachable from each source value. We refer to these settings as LLMDFA-PATHSCAN and LLMDFA-SRCSCAN, respectively. Notably, the number of prompting rounds and computational costs of LLMDFA-PATHSCAN and LLMDFA-SRCSCAN are lower than those of LLMDFA, as the latter two only reason a subset of the functions in the repository.

**Result.** Figure 6 shows the comparison results between the single-function level bug detector, multiple-function level bug detector, and REPOAUDIT. The single-function level bug detector can detect only one intra-procedural NPD bug. The key reasons are twofold. First, it only accesses the

*Listing 1.* An example of control flow facts ignored by single-function level bug detectors

```
 1.  static int __init damon_reclaim_init(void){
 2.    ctx = damon_new_ctx();
 3.    if (!ctx)
 4.        return -ENOMEM;
 5.    if (damon_select_ops(ctx, DAMON_OPS_PADDR))
 6.        return -EINVAL;
 7.    ctx->callback.after_wmarks_check = damon_reclaim_after_wmarks_check;
 8.    ctx->callback.after_aggregation = damon_reclaim_after_aggregation;
 9.    target = damon_new_target();
10.    if (!target) {
11.        damon_destroy_ctx(ctx);
12.        return -ENOMEM;
13.    }
14.    damon_add_target(ctx, target);
15.    schedule_delayed_work(&damon_reclaim_timer, 0);
16.    damon_reclaim_initialized = true;
17.    return 0;
18.  }
```

*Table 5.* The ratios of the prompting rounds (**Prompt**) and input token costs (**In_token**) of LLMDFA under two settings

| Bug Type | ID | LLMDFA-PATHSCAN | | LLMDFA-SRCSCAN | |
|---|---|---|---|---|---|
| | | **Prompt** | **In_Token** | **Prompt** | **In_Token** |
| NPD | N1 | 115.39 | 81.23 | 871.61 | 522.40 |
| | N2 | 217.00 | 120.23 | 2,839.00 | 2,087.27 |
| | N3 | 6.83 | 3.13 | 255.57 | 224.82 |
| | N4 | 115.76 | 74.91 | 778.17 | 715.23 |
| | N5 | 199.43 | 125.53 | 7,121.86 | 5,350.62 |
| UAF | U1 | 41.89 | 23.08 | 4,536.75 | 2,530.74 |
| | U2 | 24.00 | 12.29 | 50.00 | 26.26 |
| | U3 | 759.48 | 693.73 | 17.63 | 6.17 |
| | U4 | 167.50 | 95.58 | 46.60 | 25.42 |
| | U5 | 4.98 | 2.09 | 853.20 | 447.36 |
| **Average** | | **165.23** | **123.18** | **1737.04** | **1193.63** |

last function in the buggy path and lacks the calling context of the function. In NPD detection, for example, the single-function level detector cannot determine whether the parameter is null or not, which makes it fail to detect interprocedural bugs. Second, the model may overlook essential control flows within a function, which results in its inability to accurately identify the data-flow facts of a specific value, thereby achieving low recall in intra-procedural bug detection. For instance, as shown by Listing 1, the LLM ignores the error handling branch at line 6, which causes the function to return without releasing the memory object allocated by the function `damon_new_ctx()`, leading to its failure to detect this memory leak bug. With access to the full calling context, the multiple-function level bug detector is able to detect 10 bugs. This demonstrates that providing additional calling context can improve the bug detection capabilities of the LLM. However, the improvement is still limited. Due to hallucinations, the LLM may still wrongly analyze certain intra-procedural and inter-procedural dataflow facts, resulting in a significant number of FNs. In contrast, REPOAUDIT performs the path-sensitive reasoning by employing the program abstraction, which facilitates precisely discovering data-flow facts for code auditing.

Table 5 shows the comparison results between LLMDFA and REPOAUDIT. On average, the number of prompting rounds for LLMDFA-PATHSCAN is 165 times that of RE-POAUDIT, and the input token count is 123 times higher. Similarly, for LLMDFA-SRCSCAN, the number of prompting rounds is 1,737 times that of REPOAUDIT, with an input token count 1,193 times higher. It is important to note that the actual cost of LLMDFA would be even greater in practical scenarios. This significant performance gap stems from the fact that LLMDFA follows a desgin similar to compiler-based scanners that first collect all primitive data-flow facts and then correlate them to find bugs. For example, to detect the NPD bugs in the example program shown in Figure 1, LLMDFA begins with the NULL value at line 4 in the function `field2json`, analyzing the dependencies of it with all the arguments, return value, and dereference pointers (e.g., `field` at line 8) within this function. Hence, LLMDFA fails to scale to large-size projects in the real-world senarios.

## B. Comparison with Industrial Bug Detectors

**Setup and Metrics.** We choose two typical static bug detectors as the representatives of industrial tools, namely Meta INFER (Meta, 2025) and Amazon CODEGURU (Amazon, 2025). Specifically, Meta INFER is a static analysis tool from Meta. Benefiting from its sophisticated memory model (Calcagno et al., 2009), Meta INFER features its outstanding ability in memory bug detection. All three bug types in our evaluation are supported by Meta INFER. Notably, Meta INFER is only applicable to projects that can be successfully compiled. Amazon CODEGURU, as an AWS service, combines machine learning and automated reasoning to identify underlying bugs. Unlike Meta INFER, Amazon CODEGURU can directly analyze source code without compilation. As Amazon CODEGURU does not support MLK detection, we only evaluate it for the NPD and UAF detection. After running the two industrial bug detectors, we manually check the bug reports, label the TPs and FPs, and compare with the results by REPOAUDIT shown by Table 2 in Section 4.2.

**Result.** Table 3 presents the results of Meta INFER and Amazon CODEGURU. As shown by the column **Build** in Table 3, 13 projects can be successfully compiled However, five of them still cause crashes in Meta INFER, namely the projects with the IDs N3, M3, M5, U1, and U2, even after we have tried multiple versions of Infer, including v1.2.0 (latest), v1.0.0, and v0.9.0. We have reported these issues to Meta INFER, but haven't received any response yet. The failures of compiling and analyzing successfully compiled projects demonstrate the restricted applicability and instability of compilation-dependent bug detectors. Eventually, we successfully analyze eight projects with Meta INFER, obtaining a total of seven TPs and two FPs. Notably, for project `libfreenect` with the project ID N4, the five TPs generated by Infer are based on the assumption that external APIs `realloc` and `malloc` can return null pointers upon allocation failure. However, this assumption does not always hold. In our implementation, REPOAUDIT does not rely on such assumptions and instead begins its analysis with null literal values or other user-defined APIs. Lastly, although Meta INFER is known for its powerful memory model, it still fails to detect any MLK or UAF bugs discovered by REPOAUDIT. In comparison, REPOAUDIT not only supports non-compilation analysis with greater applicability but also shows stronger detection capabilities, identifying 40 TPs in total.

As shown by the last two columns in Table 3, Amazon CODEGURU does not detect any TPs upon the targeted 10 projects, while generating 18 FPs. Due to the limitations of its inherent formal reasoning techniques and machine learning models, Amazon CODEGURU can only detect certain patterns of bugs and is not robust to the various ways of writing similar buggy code.

## C. Ablation Study

**Setup and Metrics.** To evaluate the effectiveness of each technical design, we introduce three ablation variants of REPOAUDIT, namely REPOAUDIT-NOABS, REPOAUDIT-NOVAL, and REPOAUDIT-NOCACHE. Specifically, RE-POAUDIT-NOABS skips the program abstraction, i.e., removing the second step in the prompt template shown in Figure 4. REPOAUDIT-NOVAL removes the validation of the data-flow facts discovered by the explorer and also skips examining the bug reports. REPOAUDIT-NOCACHE disables the caching strategy upon the agent memory when the explorer analyzes individual functions.

**Result.** Table 6 presents the results of REPOAUDIT-NOVAL and REPOAUDIT-NOABS. Without the program abstraction, REPOAUDIT-NOABS decreases the number of TPs by 47.50%, while increasing the number of FPs by 181.82%, leading to a precision degradation to 40.38%. This decline is attributed to the complex control flows and multiple execu-

*Table 6.* The statistics of REPOAUDIT-NOABS and REPOAUDIT-NOVAL

| Bug Type | ID | REPOAUDIT-NOABS | | | REPOAUDIT-NOVAL | | |
|---|---|---|---|---|---|---|---|
| | | TP | | FP | TP | | FP |
| | | Old | New | | Old | New | |
| NPD | N1 | 1 | (3,3) | 4 | 1 | (3,3) | 2 |
| | N2 | 4 | (0,0) | 5 | 7 | (1,0) | 0 |
| | N3 | 0 | (0,0) | 6 | 1 | (1,0) | 10 |
| | N4 | 0 | (0,0) | 0 | 1 | (0,0) | 5 |
| | N5 | 1 | (2,2) | 2 | 1 | (5,4) | 4 |
| MLK | M1 | 1 | (1,1) | 2 | 1 | (2,1) | 3 |
| | M2 | 1 | (1,1) | 5 | 1 | (6,6) | 5 |
| | M3 | 1 | (0,0) | 0 | 1 | (0,0) | 0 |
| | M4 | 0 | (0,0) | 0 | 1 | (0,0) | 1 |
| | M5 | 0 | (0,0) | 0 | 1 | (0,0) | 2 |
| UAF | U1 | 1 | (0,0) | 0 | 1 | (0,0) | 0 |
| | U2 | 1 | (0,0) | 0 | 1 | (0,0) | 0 |
| | U3 | 1 | (0,0) | 1 | 1 | (0,0) | 0 |
| | U4 | 1 | (0,0) | 0 | 1 | (0,0) | 0 |
| | U5 | 0 | (1,0) | 6 | 1 | (1,0) | 6 |
| **Total** | | **13** | **8** | **31** | **21** | **19** | **38** |

*Table 7.* The computational costs of REPOAUDIT-NOCACHE. OOT: the out-of-time, indicating that REPOAUDIT-NOCACHE does not finish the code auditing in 72 hours.

| Bug Type | ID | # Prompts | | Financial($) | | Time(s) | |
|---|---|---|---|---|---|---|---|
| | | Num | Ratio | Num | Ratio | Num | Ratio |
| NPD | N1 | 436 | 3.01 | 8.65 | 2.91 | 6,316.37 | 3.12 |
| | N2 | 18 | 1.06 | 0.43 | 1.03 | 312.01 | 1.10 |
| | N3 | 175 | 1.61 | 4.12 | 1.59 | 3,098.59 | 1.77 |
| | N4 | 181 | 6.24 | 3.06 | 5.19 | 2,342.23 | 5.38 |
| | N5 | 137 | 2.17 | 4.09 | 2.37 | 2,626.21 | 2.48 |
| MLK | M1 | 459 | 2.24 | 10.49 | 2.24 | 7,070.68 | 2.42 |
| | M2 | 584 | 4.00 | 15.83 | 4.40 | 10,852.80 | 4.76 |
| | M3 | 2 | 1.00 | 0.05 | 0.95 | 34.34 | 1.00 |
| | M4 | 1 | 1.00 | 0.03 | 0.87 | 20.13 | 1.12 |
| | M5 | 200 | 5.71 | 4.88 | 5.67 | 3,635.66 | 6.06 |
| UAF | U1 | 657 | 18.25 | 14.34 | 17.93 | 11,528.96 | 19.80 |
| | U2 | 2 | 1.00 | 0.04 | 1.00 | 32.78 | 1.03 |
| | U3 | 67 | 1.40 | 2.10 | 1.62 | 1,153.46 | 1.46 |
| | U4 | 10 | 1.00 | 0.23 | 0.99 | 174.43 | 0.94 |
| | U5 | N/A | N/A | N/A | N/A | OOT | N/A |
| **Average** | | **209.21** | **3.55** | **4.88** | **3.48** | **3,514.19** | **3.75** |

tion paths created by numerous conditional branches, loop structures, and their nested combinations within functions. When analyzing such cases, the model becomes more susceptible to hallucinations, resulting in missed buggy propagation paths or incorrect identification of non-existent ones.

When the validators are disabled, the number of FPs increases to 31, causing a 245.45% increase. This rise is mainly caused by the presence of various conditional branches and jump statements, such as `if-else` and `switch`, as well as early exits in certain branches (e.g., error handling). Without the validator, the LLM may hallucinate and fail to account for these critical branches, resulting in a large number of spurious data-flow facts that do not meet feasibility requirements. Besides, the conflicts between branch conditions discovered by the LLM can also contribute to the high precision.

*Table 8.* The statistics of REPOAUDIT powered by DeepSeek R1

| Bug Type | ID | TP Old | TP New | FP | # Prompts | # Tokens Input | # Tokens Output | Financial ($) | Time (s) |
|---|---|---|---|---|---|---|---|---|---|
| NPD | N1 | 1 | (3,3) | 2 | 136 | 675,711 | 46,839 | 1.20 | 12,378.10 |
| | N2 | 7 | (2,0) | 0 | 14 | 74,918 | 6,057 | 0.14 | 1,588.65 |
| | N3 | 1 | (1,0) | 0 | 97 | 524,915 | 38,644 | 0.95 | 9,044.85 |
| | N4 | 1 | (0,0) | 0 | 7 | 29,898 | 2,288 | 0.05 | 247.99 |
| | N5 | 1 | (7,6) | 0 | 43 | 270,880 | 15,689 | 0.47 | 2,936.52 |
| MLK | M1 | 1 | (2,2) | 2 | 165 | 910,153 | 55,623 | 1.58 | 14,633.28 |
| | M2 | 1 | (9,9) | 1 | 93 | 562,470 | 35,498 | 0.98 | 5,583.13 |
| | M3 | 1 | (0,0) | 0 | 2 | 10,481 | 832 | 0.02 | 74.01 |
| | M4 | 1 | (0,0) | 0 | 1 | 5691 | 214 | 0.01 | 93.23 |
| | M5 | 1 | (0,0) | 0 | 18 | 92,573 | 8,289 | 0.18 | 1,572.31 |
| UAF | U1 | 1 | (0,0) | 0 | 50 | 255,048 | 19,494 | 0.46 | 2,444.72 |
| | U2 | 1 | (0,0) | 0 | 2 | 8900 | 704 | 0.02 | 88.76 |
| | U3 | 1 | (0,0) | 0 | 29 | 227,336 | 10,269 | 0.37 | 2,768.59 |
| | U4 | 1 | (0,0) | 0 | 11 | 53,647 | 3,617 | 0.09 | 1,997.20 |
| | U5 | 1 | (1,0) | 1 | 226 | 1,191,391 | 78,226 | 2.10 | 13,124.44 |
| Average | | | | | 59.60 | 326,267.47 | 21,485.53 | 0.57 | 4,571.72 |

*Table 9.* The statistics of REPOAUDIT powered by Claude 3.7 Sonnet

| Bug Type | ID | TP Old | TP New | FP | # Prompts | # Tokens Input | # Tokens Output | Financial ($) | Time (s) |
|---|---|---|---|---|---|---|---|---|---|
| NPD | N1 | 1 | (3,3) | 2 | 85 | 371,008 | 58,822 | 2.00 | 1,650.53 |
| | N2 | 7 | (1,0) | 0 | 15 | 98,473 | 9,015 | 0.43 | 264.68 |
| | N3 | 1 | (1,0) | 0 | 74 | 480,786 | 78,725 | 2.62 | 2,076.32 |
| | N4 | 1 | (0,0) | 0 | 15 | 62,634 | 8,822 | 0.32 | 233.96 |
| | N5 | 1 | (7,6) | 0 | 27 | 213,483 | 21,107 | 0.96 | 592.74 |
| MLK | M1 | 1 | (2,2) | 2 | 215 | 1195594 | 118735 | 5.37 | 3,510.25 |
| | M2 | 1 | (10,10) | 1 | 87 | 537290 | 61472 | 2.53 | 1,842.90 |
| | M3 | 1 | (0,0) | 0 | 2 | 10548 | 1097 | 0.05 | 46.25 |
| | M4 | 1 | (0,0) | 0 | 1 | 5741 | 1024 | 0.03 | 32.86 |
| | M5 | 1 | (0,0) | 0 | 16 | 81,970 | 9,442 | 0.39 | 402.17 |
| UAF | U1 | 1 | (0,0) | 0 | 18 | 93466 | 17159 | 0.54 | 462.51 |
| | U2 | 1 | (0,0) | 0 | 2 | 8870 | 1145 | 0.04 | 117.50 |
| | U3 | 1 | (0,0) | 0 | 82 | 605597 | 70,666 | 2.88 | 1,936.22 |
| | U4 | 1 | (0,0) | 0 | 10 | 48206 | 9823 | 0.29 | 314.60 |
| | U5 | 1 | (1,0) | 0 | 207 | 1069189 | 147422 | 5.42 | 4,387.80 |
| Average | | | | | 57.07 | 325,523.67 | 40,965.06 | 1.59 | 1,191.42 |

The results of REPOAUDIT-NOCACHE are presented in Table 7. For the project `icu`, the analysis time exceeds 72 hours and the number of prompting rounds exceeds 20,000, more than 30 times that of REPOAUDIT. For the remaining projects, on average, the number of prompting rounds increases by 3.55 times, the financial cost by 3.48 times, and the analysis time by 3.75 times compared to REPOAUDIT. Moreover, the cache hit rate is closely related to the project's size and the density of underlying graphs. In projects with intricate call graphs and dense DDGs, a single function can appear in multiple propagation paths of faulty values. For example, in the `icu` project, parsing-related functions are hit in the cache 623 times. In such cases, the caching strategy effectively reduces analysis costs and keeps the analysis time within an acceptable range.

## D. Evaluation with More Reasoning Models

Table 8, Table 9, and Table 10 show the results of REPOAUDIT with reasoning models DeepSeek R1,

Claude 3.7 Sonnet, and OpenAI o3-mini separately. In general, when integrated with reasoning models, REPOAUDIT demonstrates stronger bug detection capabilities than the one powered by Claude 3.5 Sonnet. It is able to identify all known bugs reported by existing works and can further discover additional previously unreported bugs with higher precision. This better performance is largely attributed to the reasoning model's ability to autonomously construct logical reasoning pathways, thus facilitating more precise reasoning of control-flow and data-flow facts of a program.

As shown in Table 8, REPOAUDIT powered by DeepSeek R1 successfully identifies 46 true positives, achieving a precision of 88.46%, the highest among all evaluated models. This indicates a superior capability of DeepSeek R1 in analyzing program control-flow and data-flow. In terms of cost efficiency, DeepSeek R1 incurs an average cost of $0.57 per project. However, it is also the slowest model, with an average analysis time of 4,571 seconds per project, which may be attributed to factors such as network latency and limited throughput.

*Table 10.* The statistics of REPOAUDIT powered by OpenAI o3-mini

| Bug Type | ID | TP | | FP | # Prompts | # Tokens | | Financial ($) | Time (s) |
| | | Old | New | | | Input | Output | | |
|---|---|---|---|---|---|---|---|---|---|
| NPD | N1 | 1 | (3,3) | 2 | 49 | 237,382 | 26,360 | 0.38 | 782.75 |
| | N2 | 7 | (1,0) | 0 | 17 | 104,462 | 10,286 | 0.16 | 321.19 |
| | N3 | 1 | (0,0) | 0 | 16 | 86,930 | 8,724 | 0.13 | 261.61 |
| | N4 | 1 | (0,0) | 2 | 17 | 80,877 | 8,169 | 0.12 | 299.13 |
| | N5 | 1 | (6,5) | 1 | 24 | 170,504 | 13,115 | 0.25 | 598.2588062 |
| MLK | M1 | 1 | (2,2) | 0 | 39 | 196898 | 18937 | 0.30 | 609.81 |
| | M2 | 1 | (7,7) | 2 | 46 | 303005 | 24773 | 0.44 | 1,005.86 |
| | M3 | 1 | (0,0) | 0 | 2 | 10481 | 1134 | 0.02 | 44.26 |
| | M4 | 1 | (0,0) | 0 | 1 | 5691 | 811 | 0.01 | 36.87 |
| | M5 | 1 | (0,0) | 0 | 13 | 66601 | 7672 | 0.11 | 330.96 |
| UAF | U1 | 1 | (0,0) | 0 | 24 | 128434 | 12641 | 0.20 | 266.63 |
| | U2 | 1 | (0,0) | 0 | 2 | 9422 | 958 | 0.01 | 24.08 |
| | U3 | 1 | (1,0) | 1 | 83 | 634100 | 48,528 | 0.91 | 1,176.46 |
| | U4 | 1 | (0,0) | 0 | 10 | 50966 | 4044 | 0.07 | 190.54 |
| | U5 | 1 | (1,0) | 1 | 178 | 976191 | 82933 | 1.44 | 3,427.38 |
| **Average** | | | | | 34.73 | 204,129.60 | 17,939.00 | 0.30 | 625.05 |

As shown in Table 9, REPOAUDIT powered by Claude 3.7 Sonnet also detects 46 true positives, yielding the precision of 86.79%, which is marginally below that of DeepSeek R1. Given the inherent randomness, we consider Claude 3.7 Sonnet Sonnet's program analysis capabilities to be comparable to those of DeepSeek R1. In terms of prompting rounds and token usage, Claude 3.7 Sonnet induces similar prompting rounds and input token numbers per project as DeepSeek R1. However, its output token count is 1.9 times higher. Based on the pricing policy, Claude 3.7 Sonnet is the most expensive model, resulting in the highest average financial cost of $1.59 per project.

As shown in Table 10, REPOAUDIT powered by OpenAI o3-mini demonstrates slightly lower code analysis performance, detecting 42 true positives with a precision of 82.35%. This may be due to its relatively weaker ability to capture complex code semantics, particularly in modeling data-flow, potentially resulting in both false positives and missed bugs. In terms of efficiency, OpenAI o3-mini issues significantly fewer queries per project compared to the other two models, which may also contribute to its narrower function coverage and oversight of potential data-flows. Besides, OpenAI o3-mini incurs the lowest cost per query and offers the fastest analysis speed. On average, REPOAUDIT powered by OpenAI o3-mini results in the financial cost of $0.30 and the time cost of 624.05 seconds per project.

The above statistics demonstrate that REPOAUDIT can seamlessly benefit from advances in LLM capabilities, leading to improved precision and recall in code auditing. Also, enhancements in LLM inference efficiency can further boost the overall performance of the auditing process.

## E. Evaluation with Different Temperatures

**Setup and Metrics.** In our evaluation, we set the temperature to 0.0 by default. To assess the impact of the

*Table 11.* The statistics of REPOAUDIT with different temperature settings. The column **# Reproduce** indicates the number of the reproduced bugs that are previously reported by existing works.

| Temperature | TP | FP | # Reproduce | Precision (%) | Recall (%) |
|---|---|---|---|---|---|
| 0 | 40 | 11 | 21 | 78.43 | 100 |
| 0.25 | 38 | 12 | 21 | 76.00 | 100 |
| 0.5 | 38 | 12 | 20 | 76.00 | 95.24 |
| 0.75 | 33 | 11 | 18 | 75.00 | 85.71 |
| 1.0 | 35 | 13 | 19 | 72.92 | 90.48 |

temperatures, we evaluate REPOAUDIT under four additional temperature settings: 0.25, 0.5, 0.75, and 1.0. For each setting, we record the number of true positives and false positives detected by REPOAUDIT, and compute the corresponding precision. Also, we measure the recall by investigating the proportion of reproduced bugs.

**Results.** As shown in Table 11, REPOAUDIT demonstrates stable performance across varying temperature levels, with precision remaining above 72% and recall consistently high. However, as the temperature increases to 1.0, both precision and recall decline, suggesting that greater randomness in the model's output can lead to incorrect reasoning steps, ultimately resulting in an increased number of false positives and false negatives.

## F. Examples of False Positive/Negative

We present representative a false positive and a false negative of REPOAUDIT as follows. These cases can reflect several limitations of our approach, including the model's tendency to hallucinate during complex control-flow reasoning and its insufficient understanding of implicit semantic constraints.

In Listing 2, the function `vrf_get` can return `NULL` when `name=NULL` and `vrf_id=VRF_UNKNOWN`. In the function `lib_vrf_create`, the return value of `vrf_get` is

*Listing 2.* An example of a false positive reported by REPOAUDIT due to unawareness of the **YANG** schema constraints.

```
1 struct vrf *vrf_get(vrf_id_t vrf_id, const char *name)
2 {
3     struct vrf *vrf = NULL;
4      Nothing to see, move along here
5     if (!name && vrf_id == VRF_UNKNOWN)
6         return NULL;
7     ...
8 }
```

```
1 static int lib_vrf_create(struct nb_cb_create_args *args)
2 {
3     const char *vrfname;
4     struct vrf *vrfp;
5     vrfname = yang_dnode_get_string(args->dnode, "name");
6     if (args->event != NB_EV_APPLY)
7         return NB_OK;
8     vrfp = vrf_get(VRF_UNKNOWN, vrfname);
9     SET_FLAG(vrfp->status, VRF_CONFIGURED);
10    ...
11 }
```

*Listing 3.* An example of a false negative reported by REPOAUDIT results from Claude 3.5 Sonnet overlooking the error-handling path.

```
1 struct Sass_Compiler* ADDCALL sass_make_data_compiler (struct Sass_Data_Context* data_ctx) {
2   if (data_ctx == 0) return 0;
3   Context* cpp_ctx = new Data_Context(*data_ctx);
4   return sass_prepare_context(data_ctx, cpp_ctx);
5 }
```

```
1 static Sass_Compiler* sass_prepare_context (Sass_Context* c_ctx, Context* cpp_ctx) throw() {
2   void* ctxmem = calloc(1, sizeof(struct Sass_Compiler));
3   if (ctxmem == 0) {
4     std::cerr << "Error allocating memory for context" << std::endl;
5     return 0;
6   }
7   Sass_Compiler* compiler = (struct Sass_Compiler*) ctxmem;
8   compiler->c_ctx = c_ctx;
9   return compiler;
10 }
```

assigned to the pointer `vrfp`, which is subsequently dereferenced without a null check, seemingly leading to a potential Null-Pointer-Dereference (NPD) bug. REPOAUDIT reports an NPD bug at line 9, where `vrfp->status` is accessed without checking whether `vrfp` is NULL. However, this issue is a false positive. Due to YANG schema validation, the `vrfname` variable is guaranteed to be non-NULL. Given that `vrf_get` only returns NULL when both `name=NULL` and `vrf_id=VRF_UNKNOWN`, it cannot return NULL when `vrfname!=NULL`. Hence, the dereference is safe in practice. The root cause of the false positive is that the LLMs are not aware of the fact that the return value of `yang_dnode_get_string` is never NULL.

In Listing 3, a memory object is allocated by the function `sass_make_data_compiler` and passed as the second argument to the function `sass_prepare_context`. Within `sass_prepare_context`, if `calloc` fails at line 3, `ctxmem` is set to 0, and the function returns 0 without freeing the allocated memory object `cpp_ctx` or assigning it to any other pointer, leading to a memory leak. This bug was detected by Deepseek R1 but missed by Claude 3.5 Sonnet. The latter model failed to identify the is-

sue as it did not accurately track all relevant execution paths, particularly the error-handling path in this case. In contrast, reasoning-oriented models like Deepseek R1 demonstrated superior capability in recognizing execution paths precisely, allowing REPOAUDIT to detect such memory management issues more effectively.

