# OpenReview forum: "RepoAudit: An Autonomous LLM-Agent for Repository-Level Code Auditing"
_ICML.cc/2025/Conference — ICML 2025 poster_

### Official Review · Reviewer_HHzn · 2025-02-24

**Overall Recommendation:** 3

**Summary:**

This paper presents REPOAUDIT, a system for auditing source code using large language models to identify and report software bugs. The system is designed to detect common vulnerabilities such as null pointer dereferencing, memory leak, and use after free. It utilizes a combination of parsing libraries, large language models, and a multi-step exploration process to identify potential bugs within software repositories. The system was tested against several real-world projects and compared to existing auditing methods.

**Claims And Evidence:**

The paper demonstrates an innovative approach by leveraging LLMs like Claude 3.5 Sonnet to perform data-flow analysis and control-flow validation, which significantly enhances bug detection. The system's ability to detect inter-procedural bugs is a notable strength, as demonstrated by the high number of true positive results, particularly for cross-function vulnerabilities.

**Essential References Not Discussed:**

n/a

**Experimental Designs Or Analyses:**

REPOAUDIT is both time-efficient and cost-effective. With an average analysis time of 0.44 hours per project and a low cost per bug detected, the system provides a practical and scalable solution for auditing large codebases. The comparison with traditional bug detection tools shows that REPOAUDIT is competitive, particularly in terms of precision and resource utilization.

REPOAUDIT’s performance is highly dependent on the underlying LLMs, particularly Claude 3.5 Sonnet, and the results may vary with other models. This reliance on a single model raises concerns about the system's robustness and adaptability across different LLMs or future updates. More discussion on the potential for model flexibility would strengthen the paper.

**Methods And Evaluation Criteria:**

The authors introduce two important validation mechanisms—alignment validation of data-flow facts and control flow, and feasibility validation of inter-procedural program paths. These mechanisms help ensure the accuracy of bug reports, reducing the likelihood of false positives and increasing the reliability of the tool.

While the paper focuses on three bug types (NPD, MLK, and UAF), it does not address how REPOAUDIT can handle a broader range of vulnerabilities. The generalizability of the system to other bug categories or more complex vulnerabilities is not sufficiently explored, which limits its applicability to different types of software projects.

While the paper reports false positives and true positives, there is insufficient discussion on the nature and causes of errors, especially false positives. A deeper analysis of why false positives occur and how they could be reduced or eliminated would improve the overall understanding of the limitations of REPOAUDIT.

**Other Comments Or Suggestions:**

n/a

**Other Strengths And Weaknesses:**

see above

**Questions For Authors:**

see above

**Relation To Broader Scientific Literature:**

It utilizes a combination of parsing libraries, LLMs, and a multi-step exploration process to identify potential bugs within software repositories.

**Theoretical Claims:**

n/a

---

> ### Author Rebuttal · Authors · 2025-03-30
>
> **1.Bug Customization**
>
> Please refer to the response to the first concern **Bug Customization** of [Reviewer yKxy](https://openreview.net/forum?id=TXcifVbFpG&noteId=ZFh3alkmPr).
>
> **2.Case Studies of FPs/FNs of RepoAudit**
>
> Thank you for your suggestions. We collected the following typical cases and summarized the root causes of the FPs and FNs. We will add these cases and discussions to our revision.
>
> **_Two False Positive Examples:_**
>
> **Example 1:** In the project *icu*, the function `getTimeZoneRulesAfter` contains a `goto` statement making the control flow jump to the `error` label ([code](https://anonymous.4open.science/r/cases-BBFA/example1_1.md)). RepoAudit reports an FP of a Use-After-Free (UAF) at the second uprv\_free(newTimes) at line 13 due to the hallucination of Claude-3.5-Sonnet. The model incorrectly identifies a spurious program path that leads to uprv\_free(newTimes) after `newTimes` is freed in the loop.
>
> **Example 2.** In the project `frr`, the function `vrf_get` can return NULL under certain conditions: In the function `lib_vrf_create` ([code](https://anonymous.4open.science/r/cases-BBFA/example2_2.md)), the return value of `vrf_get` ([code](https://anonymous.4open.science/r/cases-BBFA/example2_1.md)) is assigned to the pointer `vrfp`, which is subsequently dereferenced in the expression `vrfp->status` without a null check at line 13 in the function `lib_vrf_create`. RepoAudit reports it as a NPD, while it is an FP. Due to YANG schema validation, the `vrfname` variable is guaranteed to be non-NULL. Given that `vrf_get` only returns `NULL` when both `name == NULL` and `vrf_id == VRF_UNKNOWN`, it cannot return `NULL` when `vrfname != NULL`. Hence, the dereference is safe in practice. The root cause is that the LLMs are not aware of the fact that the return value of `yang_dnode_get_string` is never `NULL`.
>
> **_A False Negative Example:_**
>
> **Example 3:** In the `libsass` project, the function `sass_make_data_compiler` ([code](https://anonymous.4open.science/r/cases-BBFA/example3_1.md)) allocates a memory object and passes it as the second argument to `sass_prepare_context` ([code](https://anonymous.4open.science/r/cases-BBFA/example3_2.md)). Within `sass_prepare_context`, if `calloc` fails at line 3, `ctxmem` is set to 0, and the function returns 0 without freeing the allocated memory object `cpp_ctx` or assigning it to any other pointer, leading to a memory leak.
>
> This bug was detected by Deepseek-R1 but missed by Claude-3.5 and GPT-4o Turbo. The latter models did not accurately track all relevant execution paths, particularly the error-handling path in this case. In contrast, reasoning-oriented models like Deepseek-R1 demonstrated superior capability in recognizing execution paths, allowing RepoAudit to detect such memory management issues more effectively.
>
> **_Comparison with Existing Symbolic Tools:_**
>
> Existing symbolic code auditing tools, such as Meta Infer, can avoid the FP and FN in Example 1 and Example 3, respectively, as they symbolically enumerate all the program paths, thereby covering the program behaviors along different paths. However, they are also unable to understand the behavior of `yang_dnode_get_string` as it depends on the library function `lyd_get_value`, of which the implementation is absent in the analyzed project. Therefore, existing symbolic tools can also report the FP in Example 2.
>
> **_Future Improvements:_**
>
> - **Enhancing Model Reasoning Capabilities**: We can improve the LLM’s ability to reason by either adopting more advanced models or fine-tuning the current models for specific tasks. Fine-tuning can be tailored to particular sub-tasks like exploring feasible program paths in single functions, thus mitigating the FPs and FNs (e.g., Examples 1 and 3). Additionally, incorporating specialized training datasets focused on code analysis could further enhance the model’s accuracy in these contexts.
>
>
> - **Expanding the Tool Suite for Better Retrieval**: Another significant enhancement would involve integrating existing compilation-free analysis tools to identify all potential branches and loops within a program. This integration would offer a clearer representation of the program’s control flow structure to the model. Such an improved RAG design has the potential to significantly reduce the false positives and false negatives produced by RepoAudit, such as Examples 1 and 3.
>
> - **Adding Multi-Modal Support for Library Function Understanding**: Utilizing library documentation and other non-code material as knowledge bases of the LLMs enables RepoAudit to understand library functions better. For example, the library documentation can facilitate RepoAudit in identifying the non-null value after YANG schema validation, thereby avoiding the FP in Example 2.
>
> **3.Model Choice**
>
> Please refer to the response to the first concern **Model Choice** of [Reviewer Le35](https://openreview.net/forum?id=TXcifVbFpG&noteId=5JaOhTUECl).

---

### Official Review · Reviewer_Le35 · 2025-03-11

**Overall Recommendation:** 2

**Summary:**

This paper proposes RepoAudit, an autonomous LLM-powered code auditing framework that can compete with current academic and industry solution tools. It consists of an initiator, an explorer, and a validator, working together to efficiently analyze GitHub repositories for code quality, security vulnerabilities, and compliance issues. Based on extensive experiments, it significantly addresses existing challenges, including the inability to analyze large repositories, high computational costs, excessive false positives, and the inefficiency of traditional static analysis methods.

**Claims And Evidence:**

Yes, the claims are well supported by clear and convincing evidence.

**Essential References Not Discussed:**

No.

**Experimental Designs Or Analyses:**

The experimental design and analysis are reasonable.

**Methods And Evaluation Criteria:**

The evaluatoin criteria make sense for the problem.

**Other Comments Or Suggestions:**

A clear process flow diagram of the RepoAudit framework is needed to illustrate its overall workflow and key components. Figure 3 should be enlarged and visually enhanced, positioned as the first image to provide readers with a concrete and direct understanding of the proposed workflow.

**Other Strengths And Weaknesses:**

- I think fixing the base LLM to Claude makes the approach less general. The focus should be on the workflow rather than the inherent capabilities of the chosen LLM. Consider replacing the image that includes the Claude icon and restructuring the comparison of different LLMs in the main text instead of placing it in the appendix.

- I hope a clear table can be provided to compare current industry bug detectors and other LLM workflows, helping readers better understand the advantages of your approach.

**Questions For Authors:**

Is it possible to inject simple prompt manipulations into LLM-generated repositories to jailbreak RepoAudit and prevent it from reporting errors?

**Relation To Broader Scientific Literature:**

This paper has a broad scientific impact on software engineering in the AI era, particularly in enhancing AI's ability to automatically detect potential bugs in large-scale repositories. It has strong practical significance.

**Theoretical Claims:**

There is no theoretical claims in the paper.

---

> ### Author Rebuttal · Authors · 2025-03-30
>
> **1.Model Choice**
>
> We evaluated RepoAudit using two additional LLMs, namely DeepSeek R1 and GPT-4 Turbo, which detected 44 and 14 true bugs with precisions of 75.86% and 35.90%, respectively. More detailed statistics of RepoAudit powered by DeepSeek R1 and GPT-4 Turbo were provided in Appendix C. During the author response stage, we further evaluated RepoAudit using two more LLMs, namely Claude-3.7 and GPT-o3-mini. By scanning the experimental subjects under the same settings, RepoAudit detected 40 and 36 true bugs with precisions of 78.43% and 76.60%, respectively.
>
> As GPT-4 Turbo exhibits weaker code reasoning abilities than Claude-3.5-Sonnet and the reasoning models DeepSeek R1, Claude-3.7, and GPT-o3-mini, it would yield a weaker performance of RepoAudit. Notably, enhanced reasoning capabilities of stronger LLMs seamlessly benefit RepoAudit, demonstrating its great future potential. We will include further discussion on the performance of RepoAudit using different LLMs in the main body of the revision.
>
> **2.Comparative Table with Existing Works**
>
> We agree with your suggestion and will include the following comparative table in our revised manuscript, clearly contrasting RepoAudit against existing works:
>
> | Name              | LLM-based | Build-Free | Customizable | General Program | Repo-level |
> |-------------------|-----------|------------|--------------|------------------|------------|
> | GPTScan           | Yes       | Yes        | No           | No               | No         |
> | LLMDFA            | Yes       | Yes        | Yes          | Yes              | No         |
> | LLMSAN            | Yes       | Yes        | Yes          | Yes              | No         |
> | Meta Infer        | No        | No         | No           | Yes              | Yes        |
> | Amazon CodeGuru   | No        | Yes        | No           | Yes              | Yes        |
> | **RepoAudit**     | **Yes**   | **Yes**    | **Yes**      | **Yes**          | **Yes**    |
>
> As shown by the table, RepoAudit is the first work that supports the build-free and customizable analysis for repository-level bug detection upon general programs instead of domain-specific ones (e.g., smart contracts), supporting the security auditing of large-scale real-world software systems.
>
> **3.Workflow Figure**
>
> We will enhance the clarity and presentation of the workflow figure. Specifically, we will replace the Claude icon with a more proper icon and move the figure to page 2. Additionally, we will explicitly reference this improved diagram when describing our solution in the introduction, ensuring greater visual clarity and reader comprehension.
>
> **4.Robustness to Prompt Manipulations**
>
> Indeed, it is possible to inject (malicious) prompts into LLM-generated repositories, potentially causing RepoAudit to fail in bug detection. In the following code, we experimentally verified this by injecting several misleading natural language comments into the code. RepoAudit correctly identified a feasible buggy path and reported the confirmed bug in the original code version, but failed when analyzing the new version. Natural language comments in the code can significantly influence the inference results of LLMs, potentially misleading the model to incorrectly identify feasible paths as infeasible. An example is shown as follows:
>
> In the project `memcached`, the function `proxy_init_startfiles` allocates a memory object and assigns it to the pointer `db`. Later, if `db->buf == NULL` or `db->fname == NULL`, the function returns without freeing the pointer `db`, causing a memory leak bug.
>
> ```
>  1  struct _mcp_luafile *db = calloc(sizeof(struct _mcp_luafile), 1);
>  2  if (db == NULL) {
>  3      fprintf(stderr, "ERROR: failed to allocate memory for db\n");
>  4      return -1;
>  5  }
>  6
>  7  db->buf = calloc(db->size, 1);
>  8  if (db->buf == NULL) {
>  9      /* inject point */
> 10     fprintf(stderr, "ERROR: failed to allocate memory for db->buf\n");
> 11     return -1;
> 12 }
> 13
> 14  db->fname = strdup(p);
> 15  if (db->fname == NULL) {
> 16     /* inject point */
> 17     fprintf(stderr, "ERROR: failed to allocate memory for db->fname\n");
> 18     return -1;
> 19 }
> ```
>
> RepoAudit successfully identified this bug in the original version of the code. However, after adding the comment stating, **"This path will never be executed,"** at lines 9 and 16, RepoAudit failed to detect the bug. The key reason is that the comments mislead the LLMs to identify the buggy paths that pass the lines 9-11 and the lines 16-18 as infeasible.
>
> Lastly, it is important to note that RepoAudit is designed as a developer-oriented code auditing tool. Our threat model assumes developers do not intentionally insert misleading content into their own codebases. Our evaluation effectively demonstrates RepoAudit's practical utility in real-world scenarios. While improving LLM robustness against prompt manipulations is crucial, this aspect is orthogonal to our current research focus and beyond the scope of this work.

---

### Official Review · Reviewer_yKxy · 2025-03-13

**Overall Recommendation:** 4

**Summary:**

This paper attempts to address a key concern in LLM-based code auditing systems where repositories are too complex and big to be effectively audited by LLMs. To address this, RepoAudit explores the repository on demand by analyzing data flow relations between different sections of the repository to build a more focused context for finding bugs. It also includes a validator to check satisfiability and reduce hallucination. Overall, it shows better performance than industry-standard auditing software.

**Claims And Evidence:**

I believe the claims are clear and the evidence supports most of them. However in the introduction, one main limitation of existing systems is that using them to find new bugs requires a lot of expertise. The authors do not, to my knowledge, address this limitation in this paper. It would be nice to have a discussion about the effort required to use it in a custom repository to find bugs that may not be well known CWEs.

**Essential References Not Discussed:**

Not that I know of.

**Experimental Designs Or Analyses:**

I didn't verify the experiment results, but I believe the design is sound and the analysis makes sense. I would have appreciated more insights (mentioned earlier).

Is there a reason you did not include CodeQL as one of the baselines? It is not necessary, and suffers from much of the same issues as Infer, but it is quite popular, hence the question.

**Methods And Evaluation Criteria:**

The paper presents a good evaluation, but I would have liked to see some more insights in general, especially with the error analysis of any false positives or negatives. For instance, were there any FPs/FNs that RepoAudit missed but were captured by the other baselines? Are there any common characteristics of the bugs missed by the tool? Despite the attempts to validate the results were there any cases which slipped through?

**Other Comments Or Suggestions:**

1. Please add the number of FPs for RepoAudit in the intro, right now you only mention those for CodeGuru and Infer.
2. In section 2.2, I am unclear on what the hallucinations exhibited by the model are, and it is a bit tough to follow the explanation in Figure 2.

**Other Strengths And Weaknesses:**

I like the writing of the paper as well as the ideas presented. I personally don't see any other major weaknesses, apart from a lack of discussion on the effort required to reuse the same system for repositories written in different languages or for finding other kinds of bugs. Just a paragraph on generalizing across bugs, or coding languages (e.g. what would I need to do to use RepoAudit to find null pointer dereferences on a repository written in a different language). Actually I don't believe I saw any description of the programming language of the repositories being evaluated, so adding that is important.

**Questions For Authors:**

see above.

**Relation To Broader Scientific Literature:**

The field of bug-finding and code auditing is very actively researched, so I believe this paper is relevant to the broader scientific literature. Summarizing large data to fit the context of LLMs is a significant part of this, and using dataflow analyses to do so within repository-level code makes sense.

**Theoretical Claims:**

Didn't verify.

---

> ### Author Rebuttal · Authors · 2025-03-30
>
> **1.Customization and Expert Knowledge**
>
> What we meant in the introduction section was that in order to detect a new type of bug, a new tool often needs to be developed inside some compiler, implementing bug-specific code checking rules. This often requires substantial compiler and program analysis expertise. With our design, extending RepoAudit to a new bug type may only entail providing a textual definition of the bug and few-shot examples (please also see our response to question 4 below). That said, the current implementation of RepoAudit focuses on bugs that are caused by data-flow, which covers a wide spectrum of bug types, such as SQL injection, Cross-site Scripting, and Absolute Path Traversal. Supporting bugs that do not entail data-flow, such as concurrency bugs, requires further development. We will clarify this in the revision.
>
> **2.More Case Studies and Insights**
>
> Please refer to the response to the second concern **Case Studies of FPs/FNs of RepoAudit** of [Reviewer HHzn](https://openreview.net/forum?id=TXcifVbFpG&noteId=enl0Gy3OnD).
>
> **3.Comparison with CodeQL**
>
> Initially, we considered using CodeQL as a baseline in our evaluation. However, upon investigation, we found that CodeQL lacks built-in detectors for the three bug types targeted by RepoAudit (i.e., NPD, MLK, and UAF). Therefore, we selected Meta Infer and Amazon CodeGuru as baselines, as both provide relevant built-in analyses for the targeted memory-related bugs. We will include more justification for our baseline selection criteria in the revision.
>
> **4.Migration to Other Languages and Bug Types**
>
> Our evaluation currently focuses on C/C++ programs, but RepoAudit actually supports three additional languages: Java, Python, and Go. Supporting a new language typically requires writing a few hundred lines of Python code to implement a few primitives needed by the auditing engine, such as the caller/callee retrievers and specific program value extractors using the *tree-sitter* parsing library. Although multilingual support is not our paper’s primary contribution, we will include additional discussion on this aspect in the revised evaluation section.
>
> As we stated in our response to the first concern, RepoAudit supports various types of bugs—including those not associated with well-known CWE categories, such as domain-specific bug types. For example, in financial systems, there are often custom security policies such as *"user-sensitive data must not be logged in system outputs."* RepoAudit can handle such cases as long as the users define the form of sensitive data and logging operations as source and sink values, respectively, via several few-shot examples along with natural language descriptions. Moreover, this customization process is lightweight and incurs little manual effort. In our experiments, each bug type can be configured using no more than five few-shot examples, and the total number of words needed in the natural language prompt does not exceed 50 words.
>
> We sincerely thank the reviewer for offering valuable suggestions. We will include a more detailed discussion on the migration to other languages and bug types in the revision.

---

### Official Review · Reviewer_EkmU · 2025-03-14

**Overall Recommendation:** 2

**Summary:**

The paper presents RepoAudit, an autonomous LLM-agent designed for repository-level code auditing. RepoAudit leverages large language models to find critical bugs such as null pointer dereference, memory leak, and use-after-free. The agent efficiently scans code repositories by utilizing an agent memory system that enables on-demand exploration of relevant functions. It mimics human code auditing by focusing on path-sensitive reasoning, making it scalable and adaptable to complex repositories. By addressing the challenges of context limits and hallucinations, the system provides significant improvements over previous methods, detecting 38 true bugs in 15 real-world systems with an average cost of $2.54 per project.

**Claims And Evidence:**

1. The paper states that REPOAUDIT overcomes intrinsic LLM limitations. Controlled experiments in Sections 2.2 and 2.3 provide qualitative evidence, however, the examples are narrowly scoped. It is not entirely convincing that these mitigations will hold up consistently across more complex codebases.

2. Fairness of performance claim is unclear; specifically, the issues with INFER (such as build errors or incompatibilities) might reflect implementation or integration challenges rather than fundamental performance differences in code auditing.

**Essential References Not Discussed:**

N/A

**Experimental Designs Or Analyses:**

1. The reported failures of tools like Meta INFER may, in part, stem from integration or configuration issues rather than intrinsic limitations. This could bias the comparative results.

2. The experiments rely on a specific configuration of Claude 3.5 Sonnet (with a fixed temperature of 0.0). As LLM behavior can be sensitive to prompting parameters, it remains to be seen how robust the approach is under different settings or with other models.

**Methods And Evaluation Criteria:**

The methods and evaluation criteria are largely well-suited to the problem of repository-level code auditing.

**Other Comments Or Suggestions:**

N/A

**Other Strengths And Weaknesses:**

### Strengths

1. RepoAudit detects bugs with high precision (65.52%) and uncovers new bugs in addition to those reported by existing systems.
2. The agent’s ability to reason across different program paths, avoiding irrelevant ones, makes it effective in finding bugs like NPD, MLK, and UAF.

### Weaknesses

1. While the system mitigates hallucinations through validation, it still produces some false positives that need manual validation.
2. Though scalable, the system’s approach to managing large repositories could face performance issues with significantly larger systems or those with complex interdependencies across functions.
3. Ablation study on different prompt parameters and different LLMs.

**Questions For Authors:**

Please refer to weaknesses.

**Relation To Broader Scientific Literature:**

REPOAUDIT advances code auditing by combining established static/pointer analysis and LLM-based code understanding. This hybrid approach improves scalability and precision for repository-level analysis, addressing long-standing limitations.

**Theoretical Claims:**

The paper relies on controlled experiments and qualitative demonstrations, such as the examples of pointer handling and feasible program path exploration in Figure 2, to substantiate their assertions about the limitations of LLMs and the benefits of their approach.

---

> ### Author Rebuttal · Authors · 2025-03-30
>
> **1.Effectiveness in mitigating LLM's intrinsic limitations**
>
> Apart from the case studies in Sections 2.2 and 2.3, we evaluated RepoAudit-NoAbs that excludes program abstraction and pointer handling. The column **RepoAudit-NoAbs** in Table 5 demonstrates that this ablation decreases the number of TPs by 44.74% and increases FPs by 105%, causing precision to drop to 33.87%. Further details are available in Appendix B.3.
>
> After the paper submission, we scanned 10 more GitHub projects, which averagely have 251K LoC and 8.8K stars, indicating their high complexity and popularity. In total, RepoAudit detected 186 true bugs with the precision of 85.71%. 80 and 95 bugs have been fixed and confirmed, respectively. We will add such results to the revision.
>
> The above results strongly indicate that RepoAudit effectively mitigates the intrinsic limitations of LLMs in our context. We will change our claim from "addressing" to “mitigating” as we agree that there is still substantial room to improve.
>
>
> **2.Performance of analyzing more complex codebases**
>
> In our additional evaluation, the projects are 1.67 times the size of those in our original evaluation. Despite the increased size, RepoAudit maintained high precision (85.71%) and completed the analysis of each repository in 0.82 hours on average. This is 1.86 times the time cost of analyzing the original benchmarks. Hence, time costs are nearly linear with project sizes, showing the graceful scalability of RepoAudit.
>
> **3.Comparison with Meta Infer**
>
> Our work aims to enhance the applicability and usability of code auditing. Thus, our evaluation against Meta Infer was carefully designed to fairly reflect real-world contexts:
>
> - Given our motivation, the applicability of code auditing tools is a critical focus. In practical scenarios, build errors and compatibility issues directly affect deployment feasibility. Therefore, beyond evaluating precision, recall, and efficiency, assessing a tool's practical applicability is crucial.
>
> - The continuous evolution and variety of compilers and build systems render build-dependent tools like Meta Infer vulnerable to fundamental deployment issues—not merely integration or implementation challenges. Surveys by industry leaders highlight these fundamental obstacles [R1, R2], significantly hindering the broader deployment of the tools.
>
> In our revision, we will justify the comparison setting and explain the real-world impact of build failures and incompatibilities.
>
> > [R1] Why don't software developers use static analysis tools to find bugs? ICSE 2013
> >
> > [R2] Christakis M, Bird C. What developers want and need from program analysis: an empirical study. ASE 2016
>
> **4.Performance with alternative LLMs**
>
> Please refer to the response to the first concern **Model Choice** of [Reviewer Le35](https://openreview.net/forum?id=TXcifVbFpG&noteId=5JaOhTUECl).
>
> **5.Performance using different temperatures**
>
> Existing studies [R3, R4] on reasoning tasks suggest setting the temperature parameter to 0, thus reducing randomness in outcomes. Our work followed this practice. We will justify the setting in our revision.
>
> During the author response stage, we further evaluated RepoAudit using Claude-3.5-Sonet under different temperatures. The results are as follows.
>
> | Temp | #TP | #FP | #Reproduced | Precision (%) | Recall (%) |
> |-------------|-----|-----|--------------|----------------|-------------|
> | 0           | 38  | 20  | 21           | 65.52          | 100.00      |
> | 0.25        | 33  | 20  | 20           | 62.26          | 95.24       |
> | 0.5         | 36  | 23  | 21           | 61.02          | 100.00      |
> | 0.75        | 35  | 20  | 18           | 63.64          | 85.71       |
> | 1.0         | 33  | 24  | 18           | 57.89          | 85.71       |
>
> RepoAudit remains robust across a range of temperature settings, with precision fluctuating slightly but remaining above 57%, and recall consistently high. Here, the precision is computed by #TP/(#TP + #FP), while the recall shows the proportion of reproduced bugs. We will incorporate these detailed findings and analyses into the revision.
>
> > [R3] Satlm: Satisfiability-aided language models using declarative prompting, NeurIPS 2023
> >
> > [R4] SWE-bench: Can Language Models Resolve Real-world Github Issues? ICLR 2024
>
> **6.Manual validation of false positives**
>
> As shown by a large body of existing literature [R5], static code auditing inevitably produces false positives when detecting vulnerabilities in real-world scenarios. Manual post-validation thus remains a common practice. Notably, our evaluation shows RepoAudit surpasses the precision of SOTA industrial tools, indicating a reduction in required manual verification efforts compared to existing solutions. We will include the discussion regarding manual validation and its implications for using RepoAudit in our revision.
>
> > [R5] Mitigating false positive static analysis warnings: Progress, challenges, and opportunities, TSE 2023

---

### Decision · Program_Chairs · 2025-05-01

**Decision:**

Accept (poster)

**Comment:**

The reviewers generally agreed that the proposed approach yields strong performance on a challenging task and leverages a nontrivial approach to do so. The main concerns centered on the breadth of the evaluation (e.g. the focus on Claude) and the writing, both of which were improved/addressed in response to the reviews. In this light, we recommend accepting the paper.